# Randomized Message-Interception Smoothing: Gray-box Certificates for Graph Neural Networks

**Yan Scholten**[1], **Jan Schuchardt**[1], **Simon Geisler**[1],
**Aleksandar Bojchevski**[2] **& Stephan Günnemann**[1]
{y.scholten, j.schuchardt, s.geisler}@tum.de
bojchevski@cispa.de, s.guennemann@tum.de
[1]Dept. of Computer Science & Munich Data Science Institute, Technical University of Munich
[2]CISPA Helmholtz Center for Information Security

## Abstract

Randomized smoothing is one of the most promising frameworks for certifying the adversarial robustness of machine learning models, including Graph Neural Networks (GNNs). Yet, existing randomized smoothing certificates for GNNs are overly pessimistic since they treat the model as a black box, ignoring the underlying architecture. To remedy this, we propose novel gray-box certificates that exploit the message-passing principle of GNNs: We randomly intercept messages and carefully analyze the probability that messages from adversarially controlled nodes reach their target nodes. Compared to existing certificates, we certify robustness to much stronger adversaries that control entire nodes in the graph and can arbitrarily manipulate node features. Our certificates provide stronger guarantees for attacks at larger distances, as messages from farther-away nodes are more likely to get intercepted. We demonstrate the effectiveness of our method on various models and datasets. Since our gray-box certificates consider the underlying graph structure, we can significantly improve certifiable robustness by applying graph sparsification.[1]

## 1 Introduction

The core principle behind the majority of Graph Neural Networks (GNNs) is message passing – the representation of a node is (recursively) computed based on the representations of its neighbors (Gilmer et al., 2017). This allows for information to propagate across the graph, e.g. in a k-layer GNN the prediction for a node depends on the messages received from its k-hop neighborhood. With such models, if an adversary controls a few nodes in the graph, they can manipulate node features to craft adversarial messages that in turn change the prediction for a target node.

Such feature-based adversarial attacks are becoming significantly stronger in recent years and pose a realistic threat (Ma et al., 2020; Zou et al., 2021): Adversaries may arbitrarily manipulate features of entire nodes in their control, for example in social networks, public knowledge graphs and graphs in the financial and medical domains. Detecting such adversarial perturbations is a difficult unsolved task even beyond graphs (Carlini and Wagner, 2017), meaning such attacks may go unnoticed.

How can we limit the influence of such adversarial attacks? We introduce a simple but powerful idea: *intercept* adversarial messages. Specifically, we propose message-interception smoothing where we randomly delete edges and/or randomly ablate (mask) nodes, and analyze the probability that messages from adversarially controlled nodes reach the target nodes. By transforming any message-passing GNN into a smoothed GNN, where the prediction is the majority vote under this randomized message interception, we can provide robustness certificates (see Figure 1).

---

[1]Project page: `https://www.cs.cit.tum.de/daml/interception-smoothing`

36th Conference on Neural Information Processing Systems (NeurIPS 2022).

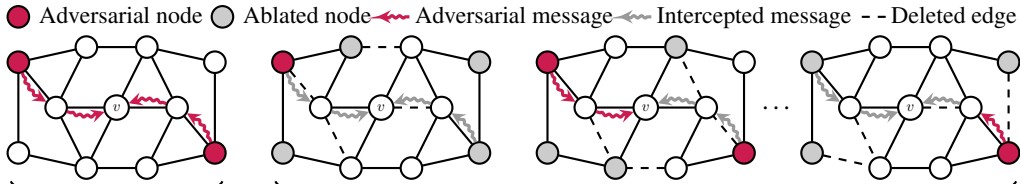

Figure 1: Randomized message-interception smoothing: We model adversaries that can arbitrarily manipulate features of multiple nodes in their control (red) to alter the predictions for a target node $v$. We intercept messages (gray) by randomly deleting edges and/or ablating (mask) all features of entire nodes. Our certificates are based on the majority vote under this randomized message interception.

Experimentally we obtain significantly better robustness guarantees compared to previous (smoothing) certificates for GNNs (compare Section 7). This improvement stems from the fact that our certificates take the underlying architecture of the classifier into account. Unlike previous randomized smoothing certificates which treat the GNN as a black-box, our certificates are *gray-box*. By making the certificate message-passing aware we partially open the black-box and obtain stronger guarantees.

Our approach is also in contrast to white-box certificates that apply only to very specific models. For example, Zügner and Günnemann (2019) only certify the GCN model (Kipf and Welling, 2017). While newly introduced GNNs require such certificates to be derived from scratch, our approach is model-agnostic and flexible enough to accommodate the large family of message-passing GNNs.

We evaluate our certificates on node classification datasets and analyze the robustness of existing GNN architectures. By applying simple graph sparsification we further increase the certifiable robustness while retaining high accuracy, as sparsification reduces the number of messages to intercept. In stark contrast to previous probabilistic smoothing-based certificates for GNNs, our certificates require only a few Monte-Carlo samples and are more efficient: For example, we can compute certificates on Cora-ML in just 17 seconds and certify robustness against much stronger adversaries than previous smoothing-based certificates (Bojchevski et al., 2020) that take up to 25 minutes.

In short, our main contributions are:

- The first gray-box smoothing-based certificates for GNNs that exploit the underlying *message-passing* principle for stronger guarantees.
- Novel randomized smoothing certificates for strong threat models where adversaries can arbitrarily manipulate features of multiple nodes in their control.

## 2 Preliminaries and Background

**Threat model.** We develop certificates for feature perturbations given *evasion* threat models. Specifically, we model adversaries that attack GNNs by entirely perturbing attributes of a few $\rho$ nodes in the graph at inference. Given an attributed graph $G = (\boldsymbol{A}, \boldsymbol{X}) \in \mathbb{G}$ encoded via adjacency matrix $\boldsymbol{A} \in \{0, 1\}^{n \times n}$ and feature matrix $\boldsymbol{X} \in \mathbb{R}^{n \times d}$ with $n$ nodes and $d$ features, we formally define the threat model of feature perturbations as a ball centered at a given graph $G = (\boldsymbol{A}, \boldsymbol{X})$:

$$B_\rho(G) \triangleq \{G' = (\boldsymbol{A}', \boldsymbol{X}') \mid \boldsymbol{A} = \boldsymbol{A}', \delta(G, G') \leq \rho\}$$

where $\delta(G, G') \triangleq \sum_{v=1}^{n} \mathbf{1}_{\boldsymbol{x}_v \neq \boldsymbol{x}'_v}$ denotes the number of nodes whose features differ in at least one dimension when comparing the clean graph $G$ and the perturbed graph $G'$. Intuitively, this means adversaries control up to $\rho$ nodes in the graph and can arbitrarily manipulate node features.

**Graph neural networks.** We design robustness certificates for GNNs that instantiate the so-called message-passing framework (Gilmer et al., 2017). The message-passing framework describes a large family of GNN architectures that are based on the local aggregation of information from neighboring nodes in the graph. To compute a new representation $\boldsymbol{h}_v^{(\ell)}$ of node $v$, each message-passing layer $\Psi^{(\ell)}$ transforms and aggregates the representations $\boldsymbol{h}_v^{(\ell-1)}$ and $\boldsymbol{h}_u^{(\ell-1)}$ of all nodes $u$ in the local neighborhood $\mathcal{N}(v) \triangleq \{u \mid \boldsymbol{A}_{uv} = 1\}$ of node $v$.

We can formally describe a message-passing layer as follows: $\boldsymbol{h}_v^{(\ell)} \triangleq \Psi_{u \in \mathcal{N}(v) \cup \{v\}}^{(\ell)} \left( \boldsymbol{h}_v^{(\ell-1)}, \boldsymbol{h}_u^{(\ell-1)} \right)$. For node classification, message-passing GNNs with $k$ GNN-layers can be described as parametrized functions $f : \mathbb{G} \to \{1, \ldots, C\}^n$ that assign each node $v$ in graph $G$ class $f_v(G) \triangleq \operatorname{argmax}_c \boldsymbol{h}_{v,c}^{(k)}$, where $\boldsymbol{h}_v^{(0)} \triangleq \boldsymbol{x}_v \in \mathbb{R}^d$ denotes the input and $\boldsymbol{h}_v^{(k)} \in \mathbb{R}^C$ the final representation of node $v$.

**Randomized smoothing.** Our robustness certificates for GNNs build upon the randomized smoothing framework (Cohen et al., 2019; Lecuyer et al., 2019): Given any base classifier $f$, for example a message-passing GNN, we can build a "smoothed" classifier $g$ that classifies randomly perturbed input samples, and then takes the "majority vote" among all predictions. The goal is to construct a smoothed classifier that behaves similar to $f$ (for example in terms of accuracy) and for which we can prove (probabilistic) robustness certificates.

Randomized ablation (Levine and Feizi, 2020b) is a smoothing-based certificate that "ablates" the input: Unlike in randomized smoothing where the input is randomly perturbed (e.g. by adding Gaussian noise to images), in randomized ablation the input is randomly masked, for example by replacing parts of the input with a special ablation token that "hides" the original information. If the perturbed input is masked for the majority of predictions, we can issue certificates for the smoothed classifier $g$.

## 3 Randomized Message-Interception Smoothing for Graph Neural Networks

The main idea of our gray-box smoothing certificates is to intercept messages from perturbed nodes by (1) deleting edges to disconnect nodes, and/or (2) ablating nodes to mask their features (cf. Figure 1).

To implement this we introduce two independent smoothing distributions $\phi_1(\boldsymbol{A})$ and $\phi_2(\boldsymbol{X})$ that randomly apply these changes to the input graph: The first smoothing distribution $\phi_1(\boldsymbol{A})$ randomly deletes edges in the adjacency matrix ($1 \to 0$) with probability $p_d$. The second smoothing distribution $\phi_2(\boldsymbol{X})$ randomly ablates all features of nodes with probability $p_a$ by replacing their feature representations with a fixed representation token $\boldsymbol{t} \in \mathbb{R}^d$ for ablated nodes. The ablation representation $\boldsymbol{t}$ is a trainable parameter of our smoothed classifier and can be optimized during training. Introducing two independent smoothing distributions is important since our base classifiers $f$ are GNNs, which behave differently under structural changes in the graph than to feature ablation of nodes in practice.

We use this message-interception smoothing distribution $\phi(G) \triangleq (\phi_1(\boldsymbol{A}), \phi_2(\boldsymbol{X}))$ to randomly sample and then classify different graphs with a message-passing GNN $f$. Finally, our smoothed classifier $g$ takes the majority vote among the predictions of $f$ for the sampled graphs $\phi(G)$. We formally describe our smoothed classifier $g$ as follows:

$$g_v(G) \triangleq \operatorname*{argmax}_{y \in \{1, \ldots, C\}} p_{v,y}(G) \qquad p_{v,y}(G) \triangleq p(f_v(\phi(G)) = y)$$

where $p_{v,y}(G)$ denotes the probability that the base GNN $f$ classifies node $v$ in graph $G$ as class $y$ under the smoothing distribution $\phi(G) = (\phi_1(\boldsymbol{A}), \phi_2(\boldsymbol{X}))$.

## 4 Provable Gray-box Robustness Certificates for Graph Neural Networks

We derive provable certificates for the smoothed classifier $g$. To this end, we develop a condition that guarantees $g_v(G) = g_v(G')$ for any graph $G' \in B_\rho(G)$: We make the worst-case assumption that adversaries alter the prediction for a target node whenever it receives at least one message from perturbed nodes. Let $E$ denote the event that at least one message from perturbed nodes reaches a target node $v$. Then the probability $\Delta \triangleq p(E)$ quantifies how much probability mass of the distribution $p_{v,y}(G)$ over classes $y$ is controlled by the worst-case adversary:

**Proposition 1.** *Given target node $v$ in graph $G$, and adversarial budget $\rho$. Let $E$ denote the event that the prediction $f_v(\phi(G))$ receives at least one message from perturbed nodes. Then the change in label probability $|p_{v,y}(G) - p_{v,y}(G')|$ is bounded by the probability $\Delta = p(E)$ for all classes $y \in \{1, \ldots, C\}$ and graphs $G'$ with $G' \in B_\rho(G)$: $|p_{v,y}(G) - p_{v,y}(G')| \leq \Delta$.*

*Proof sketch* (Proof in Appendix A). Whenever we intercept all adversarial messages, adversaries cannot alter the prediction. Thus $|p_{v,y}(G) - p_{v,y}(G')|$ is bounded by $\Delta$. $\qquad \square$

Note that we derive an upper bound on $\Delta$ in Section 5.

We first consider the special case of node ablation smoothing, discuss its relation to randomized ablation for image classifiers (Levine and Feizi, 2020b), and then we derive our provably stronger guarantees for the general case of message-interception smoothing.

**Special case of node ablation smoothing.** For the special case of node feature ablation smoothing only ($p_d = 0$), we can directly determine the probability $\Delta$ (Proof in Appendix B):

**Proposition 2.** *For node feature ablation smoothing only ($p_d = 0$), we have $\Delta = 1 - p_a^\rho$.*

In this special case, our certificates for GNNs are theoretically related to the randomized ablation certificates for image classifiers (Levine and Feizi, 2020b). We could apply their smoothing distribution to GNNs by randomly ablating features of entire nodes, instead of pixels in an image. However, their approach is specifically designed for image classifiers and comes with serious shortcomings when applied to GNNs. Notably, our robustness cetificates are provably tighter and experimentally stronger even in this special case without edge deletion smoothing ($p_d = 0$): Given that $\Delta^L$ denotes the bounding constant as defined by Levine and Feizi (2020b), we show $\Delta < \Delta^L$ in Appendix B. We carefully discuss such differences with more technical details in Appendix B. Most importantly, their certificate applied to GNNs ignores the underlying graph structure.

**General case of message-interception smoothing.** In contrast, our message-interception certificates are specifically designed for *graph-structured* data, message-passing aware, and consider the interception of messages via edge deletion as follows:

Consider a fixed target node $v$ in the graph. The formal condition for intercepting messages from a fixed target node $v$ to itself is $\phi_2(\boldsymbol{x}_v) = \boldsymbol{t}$, since we only intercept messages from the target node to the target node itself if we ablate its features. To model the interception of messages from perturbed nodes $\mathbb{B}$ other than the target node, we take the graph structure $\boldsymbol{A}$ into account: We consider all simple paths $P_{wv}^k = \{(e_1, \ldots, e_i) \mid i \leq k\}$ from perturbed nodes $w \in \mathbb{B}$ to target node $v$ of length at most $k$ (where $k$ is the number of GNN layers).[2] Intuitively, if any edge $e$ on path $p \in P_{wv}^k$ is deleted, or the features of $w$ are ablated, messages via path $p$ get intercepted. If all messages from perturbed nodes get intercepted, adversaries cannot alter the prediction for the target node (Proof in Appendix A):

**Lemma 1.** *Given a fixed target node $v$ and perturbed nodes $\mathbb{B}$ in the graph with $v \notin \mathbb{B}$. Then $f_v(\phi(G)) = f_v(\phi(G'))$ for any graph $G' \in B_\rho(G)$ if*

$$\forall w \in \mathbb{B} : \big(\forall p \in P_{wv}^k : \exists (i,j) \in p : \phi_1(\boldsymbol{A})_{ij} = 0\big) \vee (\phi_2(\boldsymbol{x}_w) = \boldsymbol{t})$$

Since k-layer message-passing GNNs aggregate information over local neighborhoods, only features of nodes in the *receptive field* affect the prediction for a target node (only via paths with a length of at most $k$ to $v$). For any perturbed node $w \in \mathbb{B}$ outside of the receptive field we have $P_{wv}^k = \emptyset$ and the message-interception condition of Lemma 1 is always fulfilled.

In practice, however, we do not know which nodes in the graph are controlled by the adversary. To account for this, we assume adversaries control nodes indicated by $\boldsymbol{\rho}_v \in \{0, 1\}^n$ that maximize the probability of the event $E(\boldsymbol{\rho}_v)$ that target node $v$ receives perturbed messages:

**Theorem 1.** *The worst-case change in label probability $|p_{v,y}(G) - p_{v,y}(G')|$ is bounded by*

$$\Delta = \max_{||\boldsymbol{\rho}_v||_0 \leq \rho} p\left(E(\boldsymbol{\rho}_v)\right)$$

*for all classes $y \in \{1, \ldots, C\}$ and any graph $G' \in B_\rho(G)$.*

Proof in Appendix A. Finally, we provide conservative robustness certificates for the smoothed classifier $g$ by exploiting that perturbed nodes are disconnected and/or ablated and cannot send messages for the majority of predictions:

**Corollary 1** (Multi-class certificate). *Given $\Delta$ as defined in Proposition 1. Then we can certify the robustness $g_v(G) = g_v(G')$ for any graph $G' \in B_\rho(G)$ if*

$$p_{v,y^*}(G) - \Delta > \max_{\tilde{y} \neq y^*} p_{v,\tilde{y}}(G) + \Delta$$

*where $y^* \triangleq g_v(G)$ denotes the majority class, and $\tilde{y}$ the follow-up (second best) class.*

Proof in Appendix A. We also provide a certificate for binary node classification in Appendix A.

---

[2]We consider simple paths (all nodes appear only once), since we only receive perturbed messages via more complex paths iff we receive perturbed messages via the simple part of the complex path.

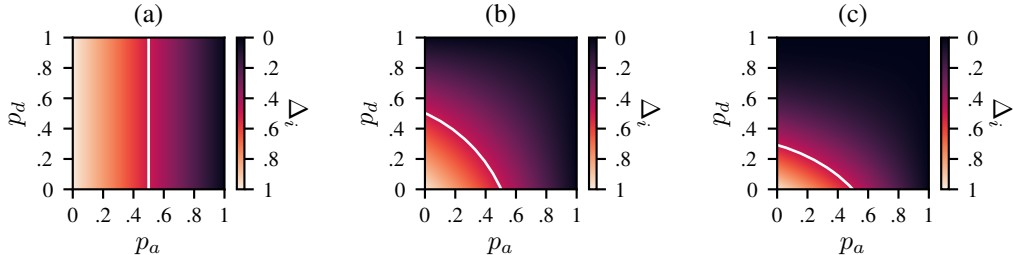

Figure 2: Single source bounding constant $\Delta_i$ for different edge deletion probabilities $p_d$ and node feature ablation probabilities $p_a$. White isolines indicate $\Delta_i = 0.5$ and separate the theoretically certifiable region ($\Delta_i < 0.5$) from the uncertifiable region ($\Delta_i \geq 0.5$). (a) For the target node, $p_d$ does not affect $\Delta_i$. (b) Direct neighbor of target node, single edge. (c) Second-hop neighbor, single path (two edges). (a-c) More distant nodes have larger theoretically certifiable regions.

## 5   Practical Interception Smoothing Certificates

Message-interception certificates constitute two challenges in practice: (1) computing the bounding constant $\Delta$ for arbitrary graphs, and (2) computing the label probabilities $p_{v,y^*}(G)$ and $p_{v,\tilde{y}}(G)$. We address the first problem by providing upper bounds on $\Delta$ (i.e. lower bounds on the certifiable robustness). For the second problem we follow existing literature and estimate the smoothed classifier.

**Lower bound on certifiable robustness.** Computing $\Delta$ of Theorem 1 poses two problems: First, finding the worst-case nodes in arbitrary graphs involves a challenging optimization over the powerset of nodes in the receptive field. Second, computing the probability $p(E(\boldsymbol{\rho}_v))$ to receive perturbed messages is challenging even for fixed $\boldsymbol{\rho}_v$, since in general, it involves evaluating the inclusion-exclusion principle (Appendix C). We can compute $\Delta$ exactly only for special cases such as small or tree-structured receptive fields (Appendix D). Notwithstanding the challenges, we provide practical upper bounds on $\Delta$. Instead of assuming a fixed $\boldsymbol{\rho}_v$, we solve both problems regarding $\Delta$ at once and directly bound the maximum over all possible $\boldsymbol{\rho}_v$ by assuming *independence* between paths. Due to Corollary 1, any upper bound on $\Delta$ result in lower bounds on the certifiable robustness.

We first derive an upper bound on $\Delta$ for a single perturbed node, and then generalize to multiple nodes. Let $E_w$ denote the event that the target node $v$ receives messages from node $w$, and $\Delta_w \triangleq p(E_w)$. Note in the special case of the target node $v = w$ we just have $\Delta_w = 1 - p_a$, since the features $\boldsymbol{x}_v$ of the target node $v$ are used for the prediction independent of any edges. For any $w \neq v$ in the receptive field we can derive the following upper bound for single sources (Proof in Appendix E):

**Theorem 2** (Single Source Multiplicative Bound). *Given target node $v$ and source node $w \neq v$ in the receptive field of a k-layer message-passing GNN $f$ with respect to $v$. Let $P_{wv}^k$ denote all simple paths from $w$ to $v$ of length at most $k$ in graph $G$. Then $\Delta_w \leq \overline{\Delta}_w$ for:*

$$\overline{\Delta}_w \triangleq \left[ 1 - \prod_{q \in P_{wv}^k} \left( 1 - (1 - p_d)^{|q|} \right) \right] (1 - p_a)$$

*where $|q|$ denotes the number of edges on the simple path $q \in P_{wv}^k$ from $w$ to $v$.*

We visualize $\Delta_w$ for different $p_d$ and $p_a$ in Figure 2. The upper bound for single sources is tight for one- and two-layer GNNs ($\Delta = \overline{\Delta}_w$), since then all paths from a single source to the target node are independent (Appendix E). The single source multiplicative bound on $\Delta_w$ can only be used to certify a radius of $\rho = 1$. For multiple nodes ($\rho > 1$), we generalize Theorem 2 as follows:

**Theorem 3** (Generalized multiplicative bound). *Assume an adversarial budget of $\rho$ nodes and let $\Delta_1, \ldots, \Delta_\rho$ denote the $\rho$ largest $\Delta_i$ for nodes $i$ in the receptive field. Then we have $\Delta \leq \overline{\Delta}_M$ for*

$$\overline{\Delta}_M \triangleq 1 - \prod_{i=1}^{\rho} (1 - \Delta_i)$$

Proof in Appendix E. Notably, the multiplicative bound is tighter than a union bound. We specifically address the approximation error in detail in Appendix F.

**Estimating the smoothed classifier in practice.** Computing the probabilities $p_{v,y^*}(G)$ and $p_{v,\tilde{y}}(G)$ exactly is challenging in practice. We instead estimate them similar to previous work by drawing Monte-Carlo samples from $\phi$ (Cohen et al., 2019; Levine and Feizi, 2020b; Bojchevski et al., 2020). We first identify the majority class $y^*$ and follow-up class $\tilde{y}$ using a few samples. We then draw more samples to estimate a lower bound $\underline{p_{v,y^*}(G)}$ on $p_{v,y^*}(G)$ and an upper bound $\overline{p_{v,\tilde{y}}(G)}$ on $p_{v,\tilde{y}}(G)$. We use the Clopper-Pearson Bernoulli confidence interval and apply Bonferroni correction to ensure that the bounds hold simultaneously with significance level $\alpha$ (with probability of at least $1-\alpha$). Moreover, our smoothed classifier abstains from predicting if $\underline{p_{v,y^*}(G)} \leq \overline{p_{v,\tilde{y}}(G)}$, meaning if the estimated probabilities are too similar. We experimentally analyze abstained predictions in Appendix H.

**Practical robustness certificates.** Finally, our robustness certificates also hold when bounding $\Delta$ and the label probabilities as the following Corollary shows (Proof in Appendix A):

**Corollary 2.** *We guarantee $g_v(G) = g_v(G')$ with probability of at least $1 - \alpha$ for any $G' \in B_\rho(G)$ if $\underline{p_{v,y^*}(G)} - \overline{\Delta} > \overline{p_{v,\tilde{y}}(G)} + \overline{\Delta}$, where $y^*$ denotes the majority class, and $\tilde{y}$ the follow-up class.*

## 6  Discussion

Our certificates require knowledge about the graph structure $A$ and can only account for structure perturbations if the perturbed adjacency matrix $A'$ is known. While adversarial edge deletion potentially increases robustness (due to less messages to intercept), adversaries could arbitrarily increase the number of messages via edge insertion. Moreover, the number of simple paths in the graph can be huge. We argue, however, that (1) graphs are usually sparse, (2) the number of paths can be reduced via sparsification, and (3) we have to compute paths only once for each graph.

**Limitations of ablation certificates.** Since the probability to receive messages from perturbed nodes increases the more nodes are adversarial, $\Delta$ is monotonously increasing in $\rho$. Thus, the certifiable radius is bounded independent of the label probabilities (uncertifiable region for $\Delta \geq 0.5$ due to Corollary 1). This bound depends on the graph structure and changes for each target node, but in the case of node feature ablation smoothing we can directly determine the bound (Proof in Appendix I):

**Proposition 3.** *Given fixed $p_a > 0$ and $p_d = 0$, it is impossible to certify a radius $\rho$ if $p_a \leq \sqrt[\rho]{0.5}$.*

This bound is only determined by the parameters of the smoothing distribution $(p_d, p_a)$ and does not depend on the base GNN $f$. The existence of an upper bound is in stark contrast to certificates whose largest certifiable radius depends on the inverse Gaussian CDF of the label probabilities (Cohen et al., 2019). Such certificates are theoretically tighter than ablation certificates: For example, if the base classifier $f$ classifies all samples from $\phi$ the same ($p_{y^*} = 1$), they would certify a radius of $\infty$, whereas the radius of ablation-based certificates is bounded. We leave the development of even stronger gray-box certificates for GNNs to future work.

**Limitations of probabilistic certificates.** Our certificates are probabilistic and hold with significance level $\alpha$. Notably, our method still yields strong guarantees for significantly smaller confidence levels (we show additional experiments for varying $\alpha$ in Appendix H). We found that $\alpha$ has just a minor effect on the certificate strength, since increasing it cannot increase the largest certifiable radius, which is theoretically bounded. Recent works also "derandomize" probabilistic certificates, that is they compute the label probabilities exactly (Levine and Feizi, 2020a, 2021). In Appendix J we propose the first derandomization technique that leverages message-passing structures. We believe future work can build upon it towards even more efficient derandomization schemes.

**Threat model extensions.** Notably, edge-deletion smoothing ($p_d > 0$) also yields guarantees for adversarial node insertion and deletion, as disconnected nodes cannot alter the prediction.[3] As discussed above, we can only evaluate such certificates with structural information, that is how inserted/deleted nodes are connected to target nodes: Given clean graphs (as in our evaluation), we know which nodes adversaries *could delete*. Given perturbed graphs, we know which nodes *could have been inserted*. Note that although we can technically extend our method to certify adversarial edge deletion, we focus on the novel problem of arbitrary feature manipulations of entire nodes since there are already certificates against edge-modification attacks (Bojchevski et al., 2020).

---

[3]We cannot certify node insertion/deletion with feature ablation smoothing, since e.g. new nodes affect the smoothed classifier independent of whether features are ablated or not (unless we delete nodes entirely).

# 7   Experimental Evaluation

We evaluate our certificates for different GNN architectures trained on node classification datasets. Our certificates work in standard transductive learning settings used throughout the literature and we report such results in Appendix H. However, combining transductive learning with an evasion threat model comes with serious shortcomings for the evaluation of certificates, since no separate test data is available. For example, we can usually achieve high accuracy by overfitting a Multi-Layer Perceptron (MLP) to labels predicted by GNNs during training. MLPs do not propagate information through the graph at test time and are robust to adversarial messages. Instead, we evaluate our certificates in semi-supervised *inductive* learning settings with hold-out test nodes:

**Experimental setup.** As labelled nodes, we draw 20 nodes per class for training and validation, and 10% of the nodes for testing. We use the labelled training nodes and all remaining unlabeled nodes as training graph, and successively insert (hold-out) validation and test nodes. We train on the training graph, optimize hyperparameters against validation nodes, assume adversaries control nodes at test time, and compute certificates for all test nodes. We also delete edges and ablate node features during training (Appendix G). We use $n_0 = 1,000$ samples for estimating the majority class, $n_1 = 3,000$ samples for certification, and set $\alpha = 0.01$. We conduct five experiments for random splits and model initializations, and report averaged results including standard deviation (shaded areas in the plots). When comparing settings (e.g. architectures), we run $1,000$ experiments for each setting and draw deletion and ablation probabilities from $[0, 1]$ for each experiment (sampling separately for training and inference). Then, we compute dominating points on the Pareto front for each setting. For brevity, we only show points with clean accuracy of at most $5\%$ below the maximally achieved performance.

**Datasets and models.** We train our models on citation datasets: Cora-ML (Bojchevski and Günnemann, 2018; McCallum et al., 2000) with 2,810 nodes, 7,981 edges and 7 classes; Citeseer (Sen et al., 2008) with 2,110 nodes, 3,668 edges and 6 classes; and PubMed (Namata et al., 2012) with 19,717 nodes, 44,324 edges and 3 classes. We implement smoothed classifiers for four architectures with two message-passing layers: Graph convolutional networks (GCN) (Kipf and Welling, 2017), graph attention networks (GAT and GATv2) (Velickovic et al., 2018; Brody et al., 2022), and soft medoid aggregation networks (SMA) (Geisler et al., 2020). More details in Appendix G. We also compute certificates for the larger graph ogbn-arxiv (Hu et al., 2020) in Appendix H.

**Evaluation metrics.** We report the classification accuracy of the smoothed classifier on the test set (*clean accuracy*), and the *certified ratio*, that is the number of test nodes whose predictions are certifiable robust for a given radius. Since all nodes have different receptive field sizes, we also divide the certifiable radius by the receptive field size. The resulting *normalized* robustness better reflects how much percentage of the "attack surface" (that is the number of nodes the adversary could attack) can be certified. Moreover, we report the area under this (normalized) certified ratio curve (*AUCRC*). For completeness, we also report the *certified accuracy* in Appendix H, that is the number of test nodes that are correctly classified (without abstaining) *and* certifiable robust for a given radius.

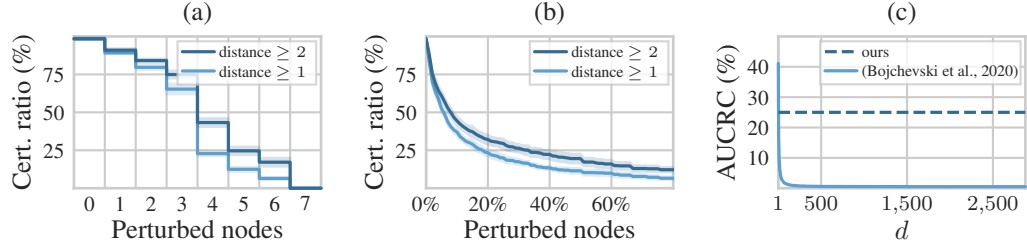

Figure 3: Smoothed GAT on Cora-ML: (a) Robustness at different distances to target nodes ($p_d$=0.31, $p_a$=0.794, with skip, ACC=0.79). (b) Robustness normalized by receptive field size ("attack surface"). (c) Naïve baseline comparison (base certificate (Bojchevski et al., 2020), $10^5$ samples, $\alpha$=0.01).

**Message-interception smoothing.** In Figure 3 (a,b) we demonstrate our certificates for specific edge deletion probabilities $p_d$ and node feature ablation probabilities $p_a$. By making our certificates message-passing aware, we can (1) certify robustness against arbitrary feature perturbations of entire nodes, (2) analyze robustness locally in the receptive fields by incorporating the "attack surface", and (3) provide stronger guarantees for attacks against nodes at larger distances to target nodes.

**First certificate for stronger adversaries.** Experimentally we obtain significantly better robustness guarantees compared to previous (smoothing-based) certificates for Graph Neural Networks. Specifically, existing certificates for GNNs only certify perturbations to a few attributes $\tilde{\rho}$ in the entire graph. Our certificates are novel as they provide guarantees for much stronger adversaries that can arbitrarily manipulate features of a multiple nodes in the graph. To compare these two approaches, consider a naïve baseline that certifies $\rho = \tilde{\rho}/d$ nodes, where $d$ is the number of attributes per node.[4] If each node in the graph had just a single feature, the number of certifiable nodes $\rho$ is high. As the number of features $d$ per node increases, however, the baseline dramatically deteriorates. In contrast, our certificates are entirely independent of the dimension $d$ and hold regardless of how high-dimensional the underlying node data might be. We demonstrate this comparison in Figure 3 (c) for the first smoothing-based certificate for GNNs (Bojchevski et al., 2020), assuming attribute deletions against second-hop nodes ($p_+$=0, $p_-$=0.6). However, the superiority of our certificate regarding robustness against all features of entire nodes holds for any other GNN certificate proposed so far.

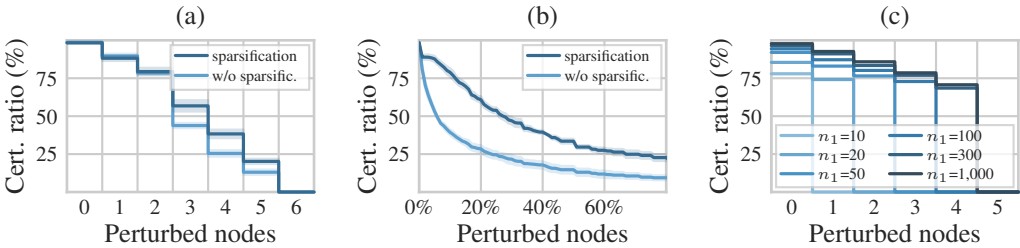

Figure 4: (a,b) Sparsification significantly improves certifiable robustness of our gray-box certificates to second-hop attacks since sparsification reduces (a) messages to intercept, and (b) receptive field sizes and thus the "attack surface" (Smoothed GAT, Cora-ML, $p_d = 0.31$, $p_a = 0.71$, with skip-connection, ACC = 0.8). (c) Our certificate with largest certifiable radius of 4 with varying samples for certification (Smoothed GAT, Cora-ML, $p_d = 0$, $p_a = 0.85$). Our certificates are more sample efficient than existing smoothing-based certificates for GNNs.

**Stronger certificates for sparser graphs.** Notably, our gray-box certificates incorporate graph structure and become stronger for sparser graphs. This is in contrast to black-box certificates that ignore the underlying message-passing principles of GNNs. We demonstrate this by applying graph sparsification, which significantly improves robustness while retaining high clean accuracy: First, sparsification reduces the number of paths in the graph and thus reduces the number of messages to intercept. Second, sparsification reduces the number of nodes in the receptive fields and thus the "attack surface", that is the number of nodes that send messages. In Figure 4 (a,b) we apply GDC preprocessing (Gasteiger et al., 2019) to the Cora-ML graph at test time. GDC preprocessing yields directed graphs and reduces the number of edges in the graph from 15,962 to 14,606 (we set the sparsification threshold of GDC to $\epsilon = 0.022$ and ignore resulting edge attributes). Interestingly, evaluating the model on the sparsified graph yields significantly higher certifiable robustness, although both approaches show high clean accuracy of 80%. Note that for the validity of our certificates we assume adversaries perturb nodes after sparsification and cannot attack the sparsification itself.

**Efficient message-interception smoothing.** Drawing Monte-Carlo samples from $\phi$ to estimate the smoothed classifier is usually the most costly part when computing smoothing-based certificates (Cohen et al., 2019). In Figure 4 (c) we show that our certificates are much more sample efficient as we do not benefit from more than a few thousand samples from $\phi$. This is in stark contrast to existing smoothing-based certificates for GNNs (Bojchevski et al., 2020). For a fair comparison, we adopt their transductive setting and compute certificates for $p_d = 0.3$ and $p_a = 0.85$. Bojchevski et al. (2020) use $10^6$ Monte-Carlo samples for certifying test nodes on Cora-ML, which takes up to 25 minutes. In contrast, our certificates saturate already for 2,000 Monte-Carlo samples in this setting, which takes only 17 seconds (preprocessing Cora-ML takes 8 additional seconds). Our gray-box certificates are significantly more sample-efficient while also providing guarantees against much stronger adversaries. We hypotheise that our certificates saturate much faster as the certifiable radius does not depend on the inverse Gaussian CDF of the label probabilities as discussed in Section 6.

---

[4]We are the first to certify such strong adversaries. Thus no baselines exist so far and we compare our method against existing certificates for GNNs using the naïve baseline we propose above.

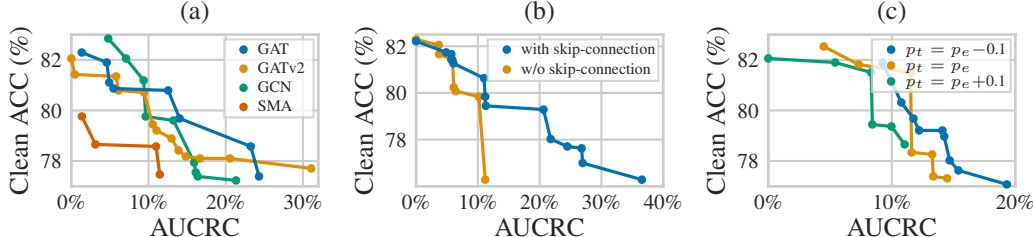

Figure 5: Second-hop attacks on Cora-ML: (a) Robustness-accuracy tradeoffs for different GNN architectures. (b) Skip-connections yield improved robustness-accuracy tradeoffs for node feature ablation smoothing. (c) Ablating less during training yields better robustness-accuracy tradeoffs (GAT).

**Different classifiers.** In Figure 5 (a) we compare robustness-accuracy tradeoffs for different GNNs against second-hop attacks. Attention-based message-passing GNNs (Velickovic et al., 2018) are dominating. We hypothesize that the degree-normalization of GCN (Kipf and Welling, 2017) may be problematic for the performance under randomized edge deletion. Our approach may promote novel message-passing architectures, specifically designed for smoothed classifiers.

**Skip-connections.** With higher node feature ablation probability, more messages from the target node itself will be intercepted, which may be detrimental for the accuracy. Assuming adversaries do not attack target nodes, we can modify the architecture for improved robustness-accuracy tradeoffs (Figure 5b). To this end, we forward the non-ablated input graph through the GNN *without edges*, and add the resulting final representation of each node to the final representation when forwarding the (ablated) graph with graph structure. We use the same weights of the base GNN, but more complex skip-connections are straightforward. Such skip-connections yield better robustness-accuracy trade-offs against second-hop attacks, but we also loose guarantees for the target node itself. To account for that, future work could deploy existing smoothing methods for features of target nodes separately: e.g., if nodes represent images, we could deploy Gaussian smoothing (Cohen et al., 2019) on node features send through the skip-connection and still obtain robustness guarantees for target nodes.

**Training-time smoothing parameters.** In Figure 5 (c) we show that ablating less during training can improve the robustness-accuracy tradeoffs. Note that only inference-time smoothing parameters determine the strength of our certificates, and the probabilities $p_d, p_a$ during training are just hyperparameters that we can optimize to improve the robustness-accuracy tradeoffs. In detail, we experiment with three different settings: Using the same ablation probabilities during training and inference ($p_t = p_e$), ablating 10% more during training ($p_t = p_e+0.1$), and ablating 10% less during training ($p_t=p_e-0.1$). Note that we use $\max(\min(p_t, 1), 0)$ to project the training-time parameters into $[0, 1]$.

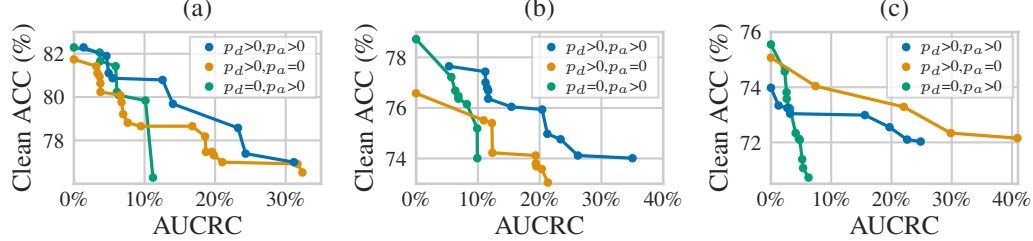

Figure 6: Robustness-accuracy tradeoffs for second-hop attacks against smoothed GAT models (without skip). Edge deletion and node ablation dominates on Cora-ML (a) and Citeseer (b). On PubMed (c), edge deletion is stronger. Lines connect dominating points on the Pareto front.

**Robustness-accuracy.** We compare robustness-accuracy tradeoffs of three different settings: (1) edge deletion and feature ablation ($p_d > 0$, $p_a > 0$), (2) edge deletion only ($p_d > 0$, $p_a = 0$), and (3) feature ablation only ($p_d = 0$, $p_a > 0$). Our experiments show that edge deletion *and* feature ablation smoothing achieves significantly better robustness-accuracy tradeoffs against attribute attacks to the *second-hop* neighborhood and dominates on Cora-ML and Citeseer (Figure 6b,c). On PubMed, edge deletion smoothing dominates. More results (e.g. with skip-connections) in Appendix H.

# 8 Related Work

**GNN robustness.** The vast majority of GNN robustness works focus on heuristic defenses, including adversarial graph detection (Zhang and Ma, 2020; Zhang et al., 2019a); architecture modifications (Brody et al., 2022; Zhang et al., 2019b); robust aggregations (Geisler et al., 2020); robust training procedures (Xu et al., 2019; Zügner and Günnemann, 2019), transfer learning (Tang et al., 2020); and graph preprocessing techniques such as edge pruning (Zhang and Zitnik, 2020; Wu et al., 2019), low-rank approximations (Entezari et al., 2020), and graph anomaly detection (Ma et al., 2021).

The effectiveness of such seemingly robust defenses on the adversarial robustness of GNNs can only be assessed against existing adversarial attacks. Heuristic defenses do not guarantee robustness, and may even be broken by stronger attacks later on (Mujkanovic et al., 2022). Instead, we are interested in robustness certificates that *provably guarantee* the stability of predictions. However, robustness certificates for GNNs are still in their infancy (Günnemann, 2022):

**Certificates for GNNs.** Most certificates for GNNs are designed for specific architectures (Zügner and Günnemann, 2020; Jin et al., 2020; Bojchevski and Günnemann, 2019; Zügner and Günnemann, 2019). Despite providing provable robustness guarantees, their applicability is limited to specific architectures. Bojchevski et al. (2020) present the first tight and efficient smoothing-based, model-agnostic certificate for graph-structured data. However, their method comes with crucial limitations: First, their method cannot certify robustness against arbitrary feature modifications of entire nodes. Second, their black-box certificate deletes edges but completely ignores the underlying *message-passing* principle. Third, their certificate requires an expensive evaluation of the smoothed classifier, which questions the practicability of their certificate beyond theoretical robustness assessments.

Randomized ablation certificates for image classifiers (Levine and Feizi, 2020b) are another approach for discrete data. Such certificates have already been applied to point cloud classifiers (Liu et al., 2021) and even for individual attribute perturbations in GNNs (Bojchevski et al., 2020). However, Bojchevski et al. (2020) show that their method outperforms such ablation certificates for individual attributes. In contrast, we propose to certify entire nodes, instead of only a few of their attributes. As already discussed, applying their ablation certificates for image classifiers directly to GNNs comes with serious shortcomings that we overcome (Section 4 and details in Appendix B).

**Gray-box certificates.** Exploiting model knowledge to derive tighter randomized smoothing certificates constitutes a widely unexplored research problem. The first works derive tighter guarantees using information about the model's gradients (Mohapatra et al., 2020; Levine et al., 2020). Recently proposed collective certificates (Schuchardt et al., 2021) incorporate knowledge about the receptive fields of GNNs. Their certificates are *orthogonal* to ours, and our certificates could lead to significant improvements in such collective settings, as adversaries cannot attack first-hop neighbors of all nodes simultaneously. Schuchardt and Günnemann (2022) propose tight gray-box certificates for models that are invariant to spatial transformations.

# 9 Conclusion

We propose novel gray-box, message-passing aware robustness certificates for GNNs against strong threat models where adversaries can arbitrarily manipulate all features of multiple nodes in the graph. The main idea of our certificates is to intercept adversarial messages by randomly deleting edges and/or masking features of entire nodes. Our certificates are significantly stronger and more sample-efficient than existing methods. Future enhancements could smooth specific edges and nodes with different probabilities, for example to intercept messages from central nodes with higher probability. Our gray-box certificates could lead to novel architectures, training techniques and graph preprocessing techniques to further strengthen the robustness of GNNs against adversarial examples.

## Acknowledgments and Disclosure of Funding

This work has been funded by the German Federal Ministry of Education and Research, the Bavarian State Ministry for Science and the Arts, and the German Research Foundation, grant GU 1409/4-1. The authors of this work take full responsibility for its content.

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
