# A Proofs Main Certificate (Section 4)

**Proposition 1.** *Given target node $v$ in graph $G$, and adversarial budget $\rho$. Let $E$ denote the event that the prediction $f_v(\phi(G))$ receives at least one message from perturbed nodes. Then the change in label probability $|p_{v,y}(G) - p_{v,y}(G')|$ is bounded by the probability $\Delta = p(E)$ for all classes $y \in \{1, \ldots, C\}$ and graphs $G'$ with $G' \in B_\rho(G)$: $|p_{v,y}(G) - p_{v,y}(G')| \leq \Delta$.*

*Proof.* For a thorough formal proof in the context of image classifiers see (Levine and Feizi, 2020b). Here, we show the statement in the context of GNNs: Consider a fixed target node $v$. We exploit that whenever we intercept all adversarial messages (i.e. nodes are disconnected or we mask out their features), the adversary cannot alter the prediction. Let $\bar{E}$ denote the event that $v$ does not receive any message from perturbed nodes. Then we have for any class $y$:

$$p(f_v(\phi(G)) = y \mid \bar{E}) = p(f_v(\phi(G')) = y \mid \bar{E})$$

since all input representations with respect to $G$ and $G'$, which affect the prediction for $v$, are the same if all perturbed nodes are ablated or disconnected (i.e. their messages are intercepted). Multiplying with $p(\bar{E})$ yields:

$$p(f_v(\phi(G)) = y \wedge \bar{E}) = p(f_v(\phi(G')) = y \wedge \bar{E}) \tag{1}$$

Following the arguments of (Levine and Feizi, 2020b):

$$
\begin{aligned}
p_{v,y}(G) - p_{v,y}(G') &\overset{(1)}{=} p(f_v(\phi(G)) = y \wedge E) + p(f_v(\phi(G)) = y \wedge \bar{E}) - p_{v,y}(G') \\
&\overset{(2)}{=} p(f_v(\phi(G)) = y \wedge E) + p(f_v(\phi(G')) = y \wedge \bar{E}) - p_{v,y}(G') \\
&\overset{(3)}{=} p(f_v(\phi(G)) = y \wedge E) - p(f_v(\phi(G')) = y \wedge E) \\
&\leq p(f_v(\phi(G)) = y \wedge E) \\
&\overset{(4)}{\leq} p(E)
\end{aligned}
$$

where (1) and (3) follow from the law of total probability, (2) is due to inserting Equation 1, and (4) follows from $p(A \cap B) \leq p(B)$ for any events $A$ and $B$.

Analogously, $p_{v,y}(G') - p_{v,y}(G) \leq p(E)$. Thus: $|p_{v,y}(G) - p_{v,y}(G')| \leq p(E) = \Delta$ $\qquad\square$

**Lemma 1.** *Given a fixed target node $v$ and perturbed nodes $\mathbb{B}$ in the graph with $v \notin \mathbb{B}$. Then $f_v(\phi(G)) = f_v(\phi(G'))$ for any graph $G' \in B_\rho(G)$ if*

$$\forall w \in \mathbb{B} : \left( \forall p \in P_{wv}^k : \exists (i,j) \in p : \phi_1(\boldsymbol{A})_{ij} = 0 \right) \vee (\phi_2(\boldsymbol{x}_w) = \boldsymbol{t})$$

*Proof.* The prediction $f_v(\phi(G))$ cannot differ from $f_v(\phi(G'))$ if for all perturbed nodes $w \in \mathbb{B}$ we have (1) $w$ is disconnected from the target node $v$, or (2) the features of $w$ are ablated. If the smoothing distribution $\phi_1$ deletes an edge $(i,j)$ (that is $\phi(\boldsymbol{A})_{ij} = 0$), the neighborhood $\mathcal{N}(j)$ changes, and thus messages from $i$ to $j$ get intercepted on all GNN layers. That is, the final hidden representation $\boldsymbol{h}_v^{(k)}$ of a target node $v$ can only be changed by some non-ablated perturbed source node $w$ if there is at least one simple path from $w$ to $v$ of length at most $k$ such that no edge on this path is deleted. $\qquad\square$

**Theorem 1.** *The worst-case change in label probability $|p_{v,y}(G) - p_{v,y}(G')|$ is bounded by*

$$\Delta = \max_{||\boldsymbol{\rho}_v||_0 \leq \rho} p\left(E(\boldsymbol{\rho}_v)\right)$$

*for all classes $y \in \{1, \ldots, C\}$ and any graph $G' \in B_\rho(G)$.*

*Proof.* Note the difference:

- $E$ denotes the event that at least one message from perturbed nodes reaches a target node $v$

- $E(\boldsymbol{\rho}_v)$ denotes the event that at least one message from nodes indicated by $\boldsymbol{\rho}_v$ reaches a target node $v$

Put differently, the maximization amounts to the additional worst-case assumption that the adversary selects those nodes whose messages have the highest chance of getting to the target node. Importantly, we have to make this additional worst-case assumption to obtain valid robustness certificates for our threat model. □

Since the probability $\Delta$ bounds the worst-case change $|p_{v,y}(G) - p_{v,y}(G')|$ for all classes $y$, we can utilize $\Delta$ to construct robustness certificates: Intuitively, $\Delta$ bounds how much probability mass of the distribution $p_{v,y}(G)$ over labels $y$ is compromised by the worst-case adversary: If an adversary cannot shift enough probability mass to change the majority class, our smoothed classifier is robust:

**Corollary 3** (Binary Certificate). *Given $\Delta$ as defined in Then we can certify the robustness $g_v(G) = g_v(G')$ for any graph $G' \in B_\rho(G)$ if*

$$p_{v,y^*}(G) - \Delta > \frac{1}{2}$$

*where $y^* \triangleq g_v(G)$ denotes the majority class predicted by smoothed classifier g.*

*Proof.* Recall that $\Delta$ bounds how much probability mass of the distribution $p_{v,y}(G)$ over $y$ is compromised by the adversary. Let $y^* \triangleq g(G)$ denote the majority class, that is $p_{v,y^*}(G) > \frac{1}{2}$ in this binary classification setting. Thus, to change the majority class, the adversary needs to shift enough probability mass from the majority class $y^*$ to the other class $1 - y^*$. This is impossible if $p_{v,y^*}(G) - \Delta > \frac{1}{2}$, meaning the adversary cannot shift enough probability mass for a successful attack. Put differently, even in the worst-case that the adversary always changes the prediction whenever adversarial messages reach the target node, the majority class cannot be altered. □

**Corollary 1** (Multi-class certificate). *Given $\Delta$ as defined in Proposition 1. Then we can certify the robustness $g_v(G) = g_v(G')$ for any graph $G' \in B_\rho(G)$ if*

$$p_{v,y^*}(G) - \Delta > \max_{\tilde{y} \neq y^*} p_{v,\tilde{y}}(G) + \Delta$$

*where $y^* \triangleq g_v(G)$ denotes the majority class, and $\tilde{y}$ the follow-up (second best) class.*

*Proof.* To prove this, we utilize the same arguments as in the binary setting above. Here, given $p_{v,y^*}(G) - \Delta > \max_{\tilde{y} \neq y^*} p_{v,\tilde{y}}(G) + \Delta$, the adversary does not control enough probability mass of $p_{v,y}(G)$ over $y$ to alter the second-best class $\tilde{y}$ into the new majority class when classifying the perturbed graph $G'$. □

**Corollary 2.** *We guarantee $g_v(G) = g_v(G')$ with probability of at least $1 - \alpha$ for any $G' \in B_\rho(G)$ if $\underline{p_{v,y^*}(G)} - \overline{\Delta} > \overline{p_{v,\tilde{y}}(G)} + \overline{\Delta}$, where $y^*$ denotes the majority class, and $\tilde{y}$ the follow-up class.*

*Proof.* We have $p_{v,y^*}(G) - \Delta \geq \underline{p_{v,y^*}(G)} - \overline{\Delta} > \overline{p_{v,\tilde{y}}(G)} + \overline{\Delta} \geq p_{v,\tilde{y}}(G) + \Delta$ due to the assumption $\underline{p_{v,y^*}(G)} - \overline{\Delta} > \overline{p_{v,\tilde{y}}(G)} + \overline{\Delta}$. The remaining claim follows from Corollary 1 and from the fact that both bounds hold with significance level $\alpha$. □

# B    Theoretical Connection to Randomized Ablation for Image Classifiers

Our gray-box certificates for GNNs are theoretically related to the randomized ablation black-box certificates for image classifiers. In this section we thoroughly analyze the differences with more technical insights and carefully discuss how our certificates go beyond theirs. Specifically, we show that our gray-box certificates yield stronger guarantees, and are provably tighter even in the special case without additional edge deletion smoothing. In the following we introduce their certificate again, discuss the differences to our certificate, and eventually prove that our guarantees are tighter.

**Randomized Ablation.**    Levine and Feizi (2020b) introduce randomized ablation for image classifiers as follows: They define the space $B(n, k) \triangleq \{M : M \in \mathcal{P}(\{1, \ldots, n\}) \wedge |M| = k\}$ of all pixel-subsets with exactly $k$ of $n$ total pixels ($\mathcal{P}$ denoting the power set here). Then, their smoothing distribution ablates all but $k$ pixels in a uniformly drawn subset $M \in B(n, k)$. They define $\Delta^L$ as the probability to *keep* (not ablate) perturbed pixels in the image under this smoothing distribution. Assuming $\rho$ perturbed pixels in an image:

$$\Delta^L = 1 - \frac{\binom{n-\rho}{k}}{\binom{n}{k}}$$

**Discussion.**    There are various ways of applying such black-box certificates for image classifiers to certify the robustness of GNNs. One way is to use them to certify threat models where adversaries control individual attributes all over the graph (Bojchevski et al., 2020). We are interested in certifying robustness to adversaries that control all features of entire nodes in the graph instead. However, applying the smoothing distribution of Levine and Feizi (2020b) for certifying robustness to our threat model (that is by ablating entire node vectors) comes with several deficiencies, as their smoothing distribution is specifically designed for image classifiers. Most importantly, applying their certificate for image classifiers to GNNs results in black-box certificates that completely ignore the underlying message-passing principle.

In contrast, we propose gray-box certificates – we partially open the black-box and consider the underlying *message-passing* principle and paths in the graph, that is $A$ and $A^2$. This comes with two crucial advantages as we show experimentally in Section 7: First, additionally deleting edges leads to significantly better robustness guarantees for attacks against more distant nodes. Second, our certificates become increasingly stronger for sparser graphs (while their certificate applied to GNNs remains unchanged as it ignores graph structure).

## B.1    Special Case of Node Feature Ablation Smoothing

Notably, our certificates are provably tighter even without edge deletion smoothing. Specifically, we formally show the difference between our $\Delta$ for node feature ablation smoothing and $\Delta^L$ of Levine and Feizi (2020b) when naively applying their approach to GNNs by randomly ablating features of entire nodes (instead of pixels in an image). Specifically, while their smoothing distribution samples exactly $k$ out of $n$ nodes not to ablate (to keep), our smoothing distribution samples $k$ out of $n$ nodes *in expectation*. This eventually leads to $\Delta < \Delta^L$. We start by characterizing our certificate for node ablation smoothing:

**Proposition 2.** *For node feature ablation smoothing only ($p_d = 0$), we have $\Delta = 1 - p_a^\rho$.*

*Proof.* Recall the definition of the probability $\Delta$: $E$ denotes the event that at least one perturbed message reaches a target node $v$, and $\Delta \triangleq p(E)$. When only ablating nodes ($p_d = 0$), all nodes are equally important for the prediction $f_v(\phi(G))$, since messages are only intercepted in the input layer, not during the message-passing itself.

We therefore do not have an optimization problem as in Theorem 1. Instead, the probability $\Delta$ to receive perturbed messages is just the probability that at least one perturbed node is not ablated. Further, the *complementary event* denotes that all $\rho$ perturbed nodes are ablated, whose probability is just $p_a^\rho$. Thus $\Delta = 1 - p_a^\rho$.    □

Moreover, the multiplicative bound is tight in the special case of node ablation smoothing:

**Proposition 4.** *For $p_d = 0$, the multiplicative bound is tight $\overline{\Delta}_M = \Delta$.*

*Proof.* We have

$$\overline{\Delta}_i \overset{(1)}{=} \left[ 1 - \prod_{q \in P_{wv}^k} \left( 1 - (1 - p_d)^{|q|} \right) \right] (1 - p_a) \overset{(2)}{=} 1 - p_a$$

where $(1)$ is by definition, and $(2)$ due to our assumption $p_d = 0$. Therefore:

$$\overline{\Delta}_M = 1 - \prod_{i=1}^{\rho} (1 - \overline{\Delta}_i) = 1 - \prod_{i=1}^{\rho} p_a = 1 - p_a^{\rho} = \Delta$$

where the first equality is due to definition again, and the last equality follows from Proposition 2. $\square$

**Proposition 5** (Tighter guarantees)**.** *Given adversarial budget $\rho > 1$. Further assume $k > 0$. Let $\Delta^L$ denote the bounding constant for the smoothing distribution proposed by Levine and Feizi (2020b). Then $\Delta < \Delta^L$.*

*Proof.* Recall that due to uniform ablation we have (compare Levine and Feizi (2020b)):

$$\Delta^L = 1 - \frac{\binom{n-\rho}{k}}{\binom{n}{k}}$$

To compare this to our $\Delta = 1 - p_a^{\rho}$ of Proposition 2, we first need to introduce $k$ and $n$. We note that $p_a$ is the probability to ablate a single node. We thus have $p_a = 1 - \frac{k}{n}$, where $\frac{k}{n}$ amounts to the probability to "keep" (not ablate) a node. In this setting, we keep $n\frac{k}{n} = k$ nodes in expectation. We therefore have:

$$\Delta = 1 - p_a^{\rho} = 1 - \left( 1 - \frac{k}{n} \right)^{\rho}$$

We observe:

$$\frac{\binom{n-\rho}{k}}{\binom{n}{k}} = \frac{(n-\rho)!(n-k)!}{n!(n-\rho-k)!} = \prod_{i=0}^{\rho-1} \frac{n-k-i}{n-i} \overset{(1)}{<} \left( \frac{n-k}{n} \right)^{\rho} = \left( 1 - \frac{k}{n} \right)^{\rho}$$

where $(1)$ is due to the mediant inequality ($\rho > 1$ and $k > 0$):

$$\forall y < x \; \forall i > 0 : \quad \frac{y-i}{x-i} < \frac{y}{x}$$

We conclude that $\Delta < \Delta^L$. $\square$

The difference decreases for larger n, but our smoothing distribution is significantly better for small graphs/receptive fields: For example, for $n = 10$ and $k = 1$ (i.e. $p_a = 0.9$), the largest certifiable radius with our method is 6, but only 4 using their certificate.

In detail, there are two ways of applying their method for image classifiers to certify robustness of GNNs against adversaries that control all features of entire nodes in the graph: by ablating all features of $k$ out of $n$ uniformly chosen nodes (1) *in the entire graph*, or (2) *locally in each receptive field*.

**Global randomized ablation.** Assume we uniformly ablate all features of $k$ out of $n$ nodes *in the entire graph*. If the number of nodes $n$ in the graph is large, the difference between $\Delta$ and $\Delta^L$ is small. Still, the resulting black-box certificates only hold globally, not locally in the receptive fields. Such certificates ignore the receptive fields, specifically that most nodes in the graph may not even be connected to the target node. For example, in the most extreme case of $A = 0$ (meaning receptive fields only consist of target nodes), their certificate applied to GNNs remains entirely unchanged due to the black-box nature. In contrast, our gray-box certificates guarantee robustness for any $\rho$ (excluding target nodes) in this case (cf. normalized robustness in Section 7).

**Local randomized ablation.** To remedy the black-box nature of their approach, one can obtain local guarantees by ablating all features of $k$ out of the $n$ nodes *locally in the receptive field* of a target node. However, our message-interception certificates are significantly tighter even without edge deletion smoothing as receptive fields are typically small. We demonstrate this in Figure 7 where our approach yields significantly stronger guarantees in practice (since Proposition 2 makes a significant difference).

Note that when applying their approach to GNNs by ablating nodes locally, one also needs to consider each receptive field individually and cannot use full-batch training/inference as usually implemented for GNNs. Our message-interception certificates are easier to implement and more efficient as we obtain local guarantees without considering and processing all receptive fields separately.

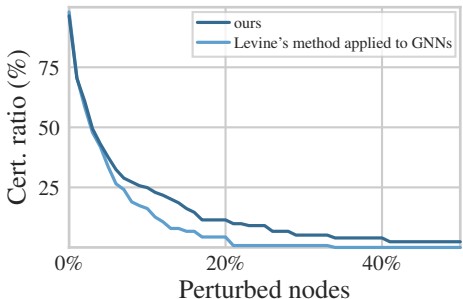

Figure 7: Given $p_a = 0.72$, we compare our certificate against the certificate proposed by Levine and Feizi (2020b) by applying their smoothing distribution for image classifiers to GNNs (distance $\geq 1$, with skip-connection). We locally choose $k = \lfloor (n-1) * p_a \rfloor$ nodes not to ablate – where $n-1$ is the number of nodes in each receptive field, excluding the target node. Our certificates are experimentally stronger even without additional edge deletion.

## C  Closed-form via Inclusion-exclusion Principle

Recall that $E(\boldsymbol{\rho}_v)$ describes the event that $v$ receives messages from *any* attacked node indicated by the adversarial budget vector $\boldsymbol{\rho}_v \in \{0, 1\}^n$. Computing the probability $p\left(E(\boldsymbol{\rho}_v)\right)$ using edge deletion probability $p_d$ and node feature ablation probability $p_a$ is challenging as it involves evaluating the inclusion-exclusion formula. We formalize this expensive closed-form solution in the following: Let $E_w$ denote the probability to receive a message from node $w$, and let $\mathcal{P}$ indicate all simple paths from any perturbed $w$ with $\boldsymbol{\rho}_v(w) = 1$ to target node $v$. Further, let $Y_i$ denote the probability to receive a message via path $i \in \mathcal{P}$. Then we have:

$$p\left(E(\boldsymbol{\rho}_v)\right) = p\left(\bigvee_{\boldsymbol{\rho}_v(w)=1} E_w\right) = p\left(\bigvee_{i \in \mathcal{P}} Y_i\right)$$

since the probability to receive a message from any attacked node equals the probability to receive a message from any path $i$ from an attacked node to the target node. We now apply the inclusion-exclusion principle:

$$p\left(\bigvee_{i \in \mathcal{P}} Y_i\right) = \sum_{k=1}^{|\mathcal{P}|}\left((-1)^{k-1} \sum_{\substack{\mathcal{I} \subseteq \mathcal{P} \\ |\mathcal{I}|=k}} p\left(\bigwedge_{i \in \mathcal{I}} Y_i\right)\right) \tag{2}$$

The remaining probability can be expressed as follows: The probability to receive messages via *all* paths indicated by $\mathcal{I}$ is the probability that (1) all edges on those paths are not deleted, and (2) the corresponding source nodes of the paths are not ablated. Therefore:

$$p\left(\bigwedge_{i \in \mathcal{I}} Y_i\right) = (1 - p_d)^a (1 - p_a)^b \tag{3}$$

where $a$ denotes the number of (unique) edges on all paths indicated by $\mathcal{I}$, and $b$ the number of (unique) source nodes of the paths indicated by $\mathcal{I}$. Note that the above derivation assumes that the target node $v$ is not controlled by the adversary. In such a case ($\boldsymbol{\rho}_v(v) = 1$), we have $p(E_v) = 1 - p_a$ (since we always receive messages from non-ablated target nodes) and:

$$p\left(E(\boldsymbol{\rho}_v)\right) = p\left(\bigvee_{i \in \mathcal{P}} Y_i \bigvee E_v\right) \qquad p\left(\bigwedge_{i \in \mathcal{I}} Y_i \bigwedge E_v\right) \overset{(1)}{=} p\left(\bigwedge_{i \in \mathcal{I}} Y_i\right) p(E_v)$$

where (1) is due to independence.

There are different ways that take additional information into account to derive faster ways of computing $p\left(E(\boldsymbol{\rho}_v)\right)$, for example by exploiting that the receptive fields are trees with the target node $v$ as root (compare Appendix D). In general, however, computing Equation 2 is expensive since we have to evaluate Equation 3 exactly $2^{|\mathcal{P}|}$ times.

# D Tree-shaped Receptive Fields

Given fixed $\boldsymbol{\rho}_v \in \{0,1\}^n$ that indicates nodes controlled by the adversary. Recall that $E(\boldsymbol{\rho}_v)$ describes the event that $v$ receives at least one messages from *any* attacked node indicated by the adversarial budget vector $\boldsymbol{\rho}_v \in \{0,1\}^n$. If the receptive field for target node $v$ is a tree, we can compute $\Delta$ of Theorem 1 exactly. Specifically, we first provide a recursive formula to compute $p\left(E(\boldsymbol{\rho}_v)\right)$ and then show that the worst-case selection of nodes by the adversary is straightforward.

We introduce the following random variables to better describe the recursion:

- Let $R_i$ denote the event that root node $i$ receives an adversarial message.
- Let $A_i$ denote the event that the features of node $i$ are ablated.
- Let $D_i$ denote the event that root $i$ receives an adversarial message via any of its adjacent subtrees $j \in \mathcal{B}$ ("branches").
- Let $B_j$ further denote the event that we receive an adversarial message via branch $j$.

The main idea is that branches in a tree are independent:

**Theorem 4.** *We start the recursion with the target node $v$ to compute $p(R_v)$ while following edges away from the node $(j, v)$ (against their direction). Then the following recursive equation computes $p\left(E(\boldsymbol{\rho}_v)\right)$ for tree-shaped receptive fields:*

$$p(R_i) \triangleq \begin{cases} 1 - p_a(1 - p(D_i)) & \text{if } \boldsymbol{\rho}_v(i) = 1 \\ p(D_i) & \text{else} \end{cases}$$

*with*

$$p(D_i) \triangleq 1 - \prod_{(j,i)} (1 - p(B_j)) \qquad p(B_j) \triangleq (1 - p_d)p(R_j)$$

*Proof.* We show the three equations consecutively:

1. For $p(R_i)$: If root $i$ is not controlled by the adversary, then the probability to receive an adversarial message is just the probability that we receive such a message via any of its adjacent subtrees, that is $p(R_i) = p(D_i)$. If root $i$ is controlled by the adversary ($\boldsymbol{\rho}_v(i) = 1$), we can exploit independence between edge deletion smoothing $\phi_1$ and node feature ablation smoothing $\phi_2$:

$$p(R_i) = p(\bar{A}_i \vee D_i) = 1 - p(A_i \wedge \bar{D}_i) \overset{(1)}{=} 1 - p(A_i)p(\bar{D}_i) = 1 - p(A_i)(1 - p(D_i))$$

   where $(1)$ is due to independence. Since the probability that we do not receive any adversarial message from root $i$ is the probability that the features of root $i$ are ablated: $p(A_i) = p_a$. We therefore have: $p(R_i) = 1 - p_a(1 - p(D_i))$.

2. For $p(D_i)$: For the probability that root $i$ receives an adversarial message via any of its adjacent branches $j \in \mathcal{B}$, we exploit independence between branches (which we can do since we have trees):

$$p(D_i) = p\left(\bigvee_{j \in \mathcal{B}} B_j\right) = 1 - p\left(\bigwedge_{j \in \mathcal{B}} \bar{B}_j\right) \overset{(1)}{=} 1 - \prod_{j \in \mathcal{B}} p(\bar{B}_j) = 1 - \prod_{j \in \mathcal{B}} (1 - p(B_j))$$

   where $(1)$ is due to independence.

3. For $p(B_j)$: The probability to receive a message via branch $j$ is the probability that the edge from branch $j$ to root $i$ is not deleted $(1 - p_d)$ times the probability that we receive a message via the next root $j$ (recursive call).

For leaves we have $\mathcal{B} = \emptyset$ and thus the product over $j \in \mathcal{B}$ is 1, that is $p(D_i) = 0$ for all leaves. $\quad\square$

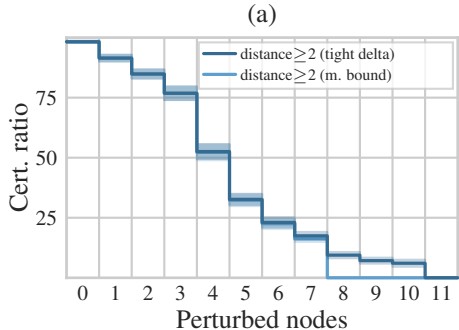
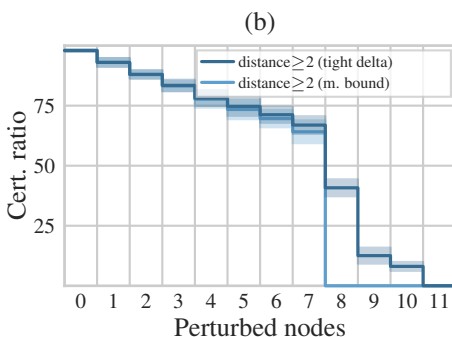

Figure 8: Comparing multiplicative bound and tight tree bound (distance at least 2). (a) Tree-certificate only for tree-shaped receptive fields. (b) Sparsifying all receptive fields into trees.

Interestingly, we can reconstruct the following special cases:

**Special case of edge deletion smoothing.** Assume $p_a = 0$. Then we directly see that $p(R_i) = 1$ if root $i$ is controlled by the adversary. This means that the adversary controls the entire sub-tree if the root node is already attacked. Put differently, the adversary does not need to control more parts of the tree to change the prediction if the adversary already controls the root.

**Special case of node feature ablation smoothing.** Assume $p_d = 0$. Then we can directly see that resolving the recursion just multiplies the node feature ablation probabilities $p_a$ and we get $p\left(E(\boldsymbol{\rho}_v)\right) = 1 - p_a^\rho$ for $\rho = ||\boldsymbol{\rho}_v||_0$. This matches the special case already discussed in Proposition 2.

**Worst-case selection of nodes.** Recall that our certificates are conservative and assume the additional worst-case that the adversary attacks those nodes in the receptive field that maximize the probability that the target node receives a message from attacked nodes (maximization in Theorem 1). This additional assumption is required to obtain valid certificates. Notably, this worst-case adversary is straightforward for trees: First, an adversary would always prefer closer nodes over more distant nodes to maximize the probability that messages are getting through. Second, an adversary would always distribute its budget over different branches to exploit independence between branches, which also maximizes the probability that messages are getting through (also compare Appendix E).

**Experiments.** We find that computing $\Delta$ tight for tree-shaped receptive fields can increase the certifiable radius in practice (compare Figure 8). Interestingly, $25\%$ of nodes in Cora-ML have receptive fields that are trees (considering 2-layer GNNs). We apply our recursive scheme above to compute tight certificates in two settings: First, we only compute tight certificates for the nodes whose receptive fields are trees. Second, we apply sparsification that successively deletes edges in the graph until the receptive fields of all test nodes are trees. In detail, we train GAT models on Cora-ML and apply sparsification at test time. We use the skip-connection, train with $p_a = 0.68$, $p_d = 0.02$ and compute certificates with $p_a = 0.79$, $p_d = 0.36$. Without sparsification we achieve clean accuracies of $79\%$ on average, and $77\%$ when applying sparsification at test time.

In practice, we found that the gain in computing $\Delta$ exactly may be rather small, as adversaries typically distribute their budget to different branches to increase the probability that their messages arrive. This means adversaries maximize independencies between edges. In other words, the multiplicative bound is already quite strong in practice, and specifically tight until the degree of the node (given that each first-hop neighbor has at least one child).

# E   Proofs of Section 5

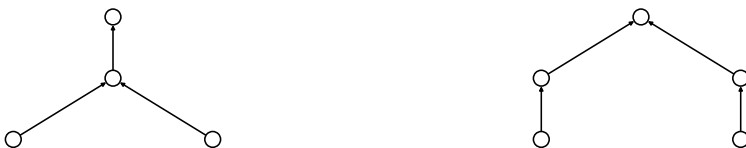

Figure 9: Visualization of two dependent (left) and independent paths (right). When randomly deleting edges with the same edge deletion probability $p_d$, the probability that all messages from both source nodes are intercepted is lower when the paths are independent (more possibilities for the message to get through).

We first prove a more general claim that we can use to prove the multiplicative bounds of Theorem 2 and Theorem 3. Let $X_i$ denote the event that target node $v$ receives a message via any path $s$ in a set of paths $S_i$ such that all paths start at an arbitrary source node and end at target node $v$. Intuitively, it is more likely to receive at least one messages via $S_i$ *and* one message via $S_j$ when there are shared edges, compared to when we assume their paths were independent. Put differently, the probability that all messages from all paths are intercepted is higher when paths are dependent (cf. Figure 9). More formally:

**Theorem 5.** *For two arbitrary sets $S_i$ and $S_j$ of simple paths with the same target node $v$ we have*

$$p(\overline{X}_i)p(\overline{X}_j) \le p(\overline{X}_i \wedge \overline{X}_j)$$

*under the smoothing distribution $\phi_1$ for edge deletion.*

*Proof.* We are interested in the probability that all messages via all paths are intercepted. Consider the following two possibilities:

1. The paths in $S_i$ and the paths in $S_j$ are (pairwise) independent, meaning there are no edges that appear on both - on a path $s_i \in S_i$ and on a path $s_j \in S_j$.

   In this case we have $p(\overline{X}_i \wedge \overline{X}_j) = p(\overline{X}_i)p(\overline{X}_j)$ due to independence.

2. Consider the scenario where there are at least two dependent paths that share a common edge. If we assume they were independent, there would be more possibilities how a message can get through than there actually are. In other words, assuming independence results in lower probability that all messages via both sets get intercepted.

   Thus $p(\overline{X}_i)p(\overline{X}_j) < p(\overline{X}_i \wedge \overline{X}_j)$. $\qquad\qquad\square$

Consider the following definition of positively associated random variables (Esary et al., 1967).

**Definition 1.** *We call a random vector $\boldsymbol{x} = (X_1, \ldots, X_n)$ positively associated if*

$$Cov(\phi(\boldsymbol{x}), \psi(\boldsymbol{x}))) \ge 0$$

*for all non-decreasing, element-wise functions $\phi$, $\psi$ such that second moments of $\psi(\boldsymbol{x})$ and $\phi(\boldsymbol{y})$ exist.*

The concept of positively associated random variables is for example used in physical statistics (Goldstein and Wiroonsri, 2018). We can use this concept here to prove multiplicative bounds:

**Corollary 4.** *The random vector $\boldsymbol{x} = (X_1, \ldots, X_n)$ is positively associated.*

*Proof.* Due to Theorem 5 we have $p(\overline{X}_i)p(\overline{X}_j) \le p(\overline{X}_i \wedge \overline{X}_j)$ and thus

$$\Rightarrow \mathbb{E}[\overline{X}_i]\mathbb{E}[\overline{X}_j] \le \mathbb{E}[\overline{X}_i\overline{X}_j]$$
$$\Rightarrow \mathbb{E}[\overline{X}_i\overline{X}_j] - \mathbb{E}[\overline{X}_i]E[\overline{X}_j] \ge 0$$
$$\Rightarrow Cov(\overline{X}_i, \overline{X}_j) \ge 0$$

since $\overline{X}_i$ and $\overline{X}_j$ are binary random variables.

Thus, the elements of the covariance matrix are non-negative: $Cov(\bar{\boldsymbol{x}}, \bar{\boldsymbol{x}}) \ge 0$ (variance is always non-negative). According to Theorem 4.2 in Esary et al. (1967), $\bar{\boldsymbol{x}}$ is positively associated. Since $\bar{\boldsymbol{x}}$ is positively associated, it follows from (BP1) in Esary et al. (1967) that $\boldsymbol{x}$ is positively associated. $\quad\square$

**Proposition 6.** *Given random variables $X_i$ as defined above. Then:*

$$1 - p\left(\bigwedge_{i=1}^{n} \overline{X}_i\right) \leq 1 - \prod_{i=1}^{n} p\left(\overline{X}_i\right)$$

*Proof.* Since $x$ and $\bar{x}$ are positively associated random variables, we can use Theorem 4.1 in (Esary et al., 1967) and conclude that

$$p\left(\bigwedge_{i=1}^{n} \overline{X}_i\right) \geq \prod_{i=1}^{n} p\left(\overline{X}_i\right) \Leftrightarrow 1 - p\left(\bigwedge_{i=1}^{n} \overline{X}_i\right) \leq 1 - \prod_{i=1}^{n} p\left(\overline{X}_i\right)$$

$\square$

**Theorem 2** (Single Source Multiplicative Bound). *Given target node $v$ and source node $w \neq v$ in the receptive field of a k-layer message-passing GNN $f$ with respect to $v$. Let $P_{wv}^k$ denote all simple paths from $w$ to $v$ of length at most $k$ in graph $G$. Then $\Delta_w \leq \overline{\Delta}_w$ for:*

$$\overline{\Delta}_w \triangleq \left[1 - \prod_{q \in P_{wv}^k} \left(1 - (1 - p_d)^{|q|}\right)\right](1 - p_a)$$

*where $|q|$ denotes the number of edges on the simple path $q \in P_{wv}^k$ from $w$ to $v$.*

*Proof.* Note in the special case of the target node $v = w$ we just have $\Delta_w = 1 - p_a$, since the features $x_v$ of the target node $v$ are used for the prediction independent of any edges.

For any $w \neq v$ in the receptive field: Let $E_w$ denote the event that the target node $v$ receives messages from node $w$, and $\Delta_w \triangleq p(E_w)$. We further introduce $A_w$ for the event that the features of node $w$ are ablated, and $D_w$ for the event that $v$ receives at least one messages from $w$. Then we have:

$$\Delta_w = p(E_w) = p(\bar{A}_w \wedge D_w) \stackrel{(1)}{=} p(\bar{A}_w)p(D_w) = (1 - p_a)p(D_w)$$

where (1) holds since the two smoothing distributions for node feature ablation and edge deletion are independent. We continue with $p(D_w)$. Therefore, recall that $\mathcal{P} \triangleq \mathcal{P}_{wv}^k$ denotes the set of simple paths from $w$ to $v$. Further, let $p(q)$ for simple path $q \in \mathcal{P}$ denote the probability that $v$ receives a message via path $q$. Clearly, a message "arrives" only via path $q$ if none of the edges on that path is deleted, that is when the node is connected via path $q$. Since the deletion of edges is independent, $p(q) = (1 - p_d)^{|q|}$, where $|q|$ denotes the number of edges on the simple path $q$. We derive:

$$p(D_i) = p\left(\bigvee_{q \in \mathcal{P}} q\right) = 1 - p\left(\bigwedge_{q \in \mathcal{P}} \bar{q}\right)$$

We can use positive association to conclude

$$1 - p\left(\bigwedge_{q \in \mathcal{P}} \bar{q}\right) \stackrel{(1)}{\leq} 1 - \prod_{q \in \mathcal{P}} p(\bar{q})$$

where (1) follows from Proposition 6. Finally, we resolve the remaining terms:

$$1 - \prod_{q \in \mathcal{P}} p(\bar{q}) = 1 - \prod_{q \in \mathcal{P}} (1 - p(q)) = 1 - \prod_{q \in \mathcal{P}} \left(1 - (1 - p_d)^{|q|}\right)$$

Due to (1) above, we finally get $\Delta_w \leq \overline{\Delta}_w$, where the inequality becomes an equality if all paths are independent (that is the paths do not share edges). $\square$

**Proposition 7.** *We have $\Delta_w = \overline{\Delta}_w$ for $\ell$-layer GNNs with $\ell \leq 2$.*

*Proof.* For $\ell$-layer GNNs with $\ell \leq 2$, all paths from a single source to the target node are independent. $\square$

**Theorem 3** (Generalized multiplicative bound). *Assume an adversarial budget of $\rho$ nodes and let $\Delta_1, \ldots, \Delta_\rho$ denote the $\rho$ largest $\Delta_i$ for nodes $i$ in the receptive field. Then we have $\Delta \leq \overline{\Delta}_M$ for*

$$\overline{\Delta}_M \triangleq 1 - \prod_{i=1}^{\rho} (1 - \Delta_i)$$

*Proof.* We recall from Theorem 1:

$$\Delta = \max_{||\boldsymbol{\rho}_v||_1 \leq \rho} p\left(E(\boldsymbol{\rho}_v)\right)$$

where $E(\boldsymbol{\rho}_v)$ describes the event that target node $v$ receives *messages* from any attacked node indicated by $\boldsymbol{\rho}_v$. Recall that $E_w$ denotes the event that the prediction for target node $v$ is based on information of node $w$ in the receptive field. We further have $\Delta_w \triangleq p(E_w)$. Then:

$$p\left(E(\boldsymbol{\rho}_v)\right) = p\left(\bigvee_{\boldsymbol{\rho}_v(w)=1} E_w\right) = 1 - p\left(\bigwedge_{\boldsymbol{\rho}_v(w)=1} \bar{E}_w\right)$$

where we can apply Proposition 6 and use the assumption that paths from several source nodes to the target were independent to obtain an upper bound:

$$1 - p\left(\bigwedge_{\boldsymbol{\rho}_v(w)=1} \bar{E}_w\right) \leq 1 - \prod_{\boldsymbol{\rho}_v(w)=1} p\left(\bar{E}_w\right)$$

Further resolving the terms yields:

$$1 - \prod_{\boldsymbol{\rho}_v(w)=1} p\left(\bar{E}_w\right) = 1 - \prod_{\boldsymbol{\rho}_v(w)=1} (1 - p\left(E_w\right)) = 1 - \prod_{\boldsymbol{\rho}_v(w)=1} (1 - \Delta_w)$$

Since the above equations hold for any fixed $\boldsymbol{\rho}_v$:

$$\Delta = \max_{||\boldsymbol{\rho}_v||_1 \leq \rho} p\left(E(\boldsymbol{\rho}_v)\right) \leq \max_{||\boldsymbol{\rho}_v||_1 \leq \rho} 1 - \prod_{\boldsymbol{\rho}_v(w)=1} (1 - \Delta_w)$$

Assume we have ordered $\Delta_w$ so that $\Delta_i \geq \Delta_{i+1}$ for all $i \in \{1, \ldots, \rho\}$. Then:

$$\max_{||\boldsymbol{\rho}_v||_1 \leq \rho} 1 - \prod_{\boldsymbol{\rho}_v(w)=1} (1 - \Delta_w) = 1 - \prod_{i=1}^{\rho} (1 - \Delta_i) = \overline{\Delta}_M$$

$\square$

Note that instead of $\Delta_w$ we can alternatively use upper bounds $\overline{\Delta}_w$, which yields an even looser upper bound on $\Delta$ since

$$1 - \prod_{i=1}^{\rho} (1 - \Delta_i) \leq 1 - \prod_{i=1}^{\rho} \left(1 - \overline{\Delta}_i\right)$$

## F Approximation Error

Notably, the multiplicative bound derived above is tighter than the following union bound:

**Proposition 8** (Union Bound)**.** *Given monotonously decreasing $\Delta_i$ such that $\Delta_i \geq \Delta_{i+1}$. Then we have $\Delta \leq \overline{\Delta}_U$ for*

$$\overline{\Delta}_U \triangleq \sum_{i=1}^{\rho} \Delta_i$$

*Proof.*

$$p\left(E(\boldsymbol{\rho}_v)\right) = p\left(\bigvee_{\boldsymbol{\rho}_v(w)=1} E_w\right) \leq \sum_{\boldsymbol{\rho}_v(w)=1} p\left(E_w\right) = \sum_{\boldsymbol{\rho}_v(w)=1} \Delta_w$$

$$\Delta = \max_{||\boldsymbol{\rho}_v||_1 \leq \rho} p\left(E(\boldsymbol{\rho}_v)\right) \leq \max_{||\boldsymbol{\rho}_v||_1 \leq \rho} \sum_{\boldsymbol{\rho}_v(w)=1} p\left(E_w\right) = \sum_{i=1}^{\rho} \Delta_i$$

$\square$

The union bound is quite loose, not a probability and can even grow larger than 1. We show the difference in practice Figure 10 (a). We also discuss the approximation error between the upper bounds $\overline{\Delta}_U, \overline{\Delta}_M$ and the tight $\Delta$ for the following constructed example where all paths are dependent: We assume a setting where an adversary attacks only second-hop neighbors that are connected to the target node via the same direct neighbor of the target node. With $p_a = 0$ we have $\Delta = (1-p_d)(1-p_d^\rho)$ since we only receive a message if the bottleneck edge is not ablated, and at least one edge of the attacked second-hop nodes is not ablated (which is the complementary probability of all second-hop edges are ablated). In this constructed case, all paths are dependent as they share the bottleneck edge. We show how the upper bounds compare to the tight $\Delta$ for different edge deletion probabilities $p_d$ in Figure 10 (b). Note that the example is constructed and worst-case adversaries aim at maximizing independencies by choosing nodes without bottleneck edges (in which case the multiplicative bound is a strong bound in practice).

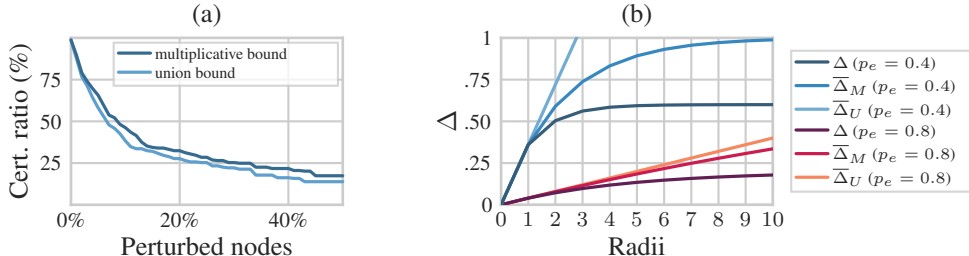

Figure 10: (a) Multiplicative bound is tighter than union bound and provides stronger guarantees (Smoothed GAT model on Cora-ML with $p_a = 0.85, p_d = 0$). (b) Constructed example: All path share the same bottleneck edge: Comparing the tight $\Delta$ against the union bound $\overline{\Delta}_U$ and the multiplicative bound $\overline{\Delta}_M$ for different edge deletion probabilities $p_d$. The multiplicative bound is tighter than the union bound, which can grow larger than 1.

# G   Hyperparameters

We implement certificates for directed and undirected graphs. For our main experiments (Section 7), however, we follow the standard procedure and prepocess all graphs into undirected graphs, only consider the largest connected component, and binarize node features. We compute simple paths using a modified depth first search. All datasets are included in PyTorch Geometric (Fey and Lenssen, 2019).[5] We train models full-batch using Adam (learning rate = 0.001, $\beta_1 = 0.9$, $\beta_2 = 0.999$, $\epsilon = 10^{-08}$, weight decay = $5 * 10^{-04}$) for 1,000 epochs with early stopping after 50 epochs. We use a dropout of $0.8$ on the feature matrix $X$ and on the attention coefficients. During training, we sample a different graph from $\phi(G)$ each epoch. Each sampled graph contains nodes with features replaced by the ablation representation $t$. We implement $t$ as a parameter of our models: We initialize $t$ using Xavier initialization and we optimize $t$ as we optimize the GNN weights during training. We implement all models for two message-passing layers. We use 8 heads and 8 hidden channels for GAT and GATv2 (Velickovic et al., 2018; Brody et al., 2022); 64 hidden channels for GCN (Kipf and Welling, 2017); and we use $k = 64$ and temperature=1.0 for SMA (Geisler et al., 2021). We use the ReLU activation function for the skip-connection. For GDC sparsification, we set the sparsification threshold of GDC to $\epsilon = 0.022$, and ignore edge attributes resulting from GDC preprocessing.

**Training-time smoothing parameters.** We also delete edges and ablate node features during training (using different probabilities $p_d$ and $p_a$ during training and inference). Specifically, we train models presented in Section 7 as follows: In Figure 3 (a,b) we show results for $p_d = 0.01, p_a = 0.6$ during training (and $p_d = 0.31, p_a = 0.794$ during inference and certification). In Figure 4 (a,b) we use $p_d = 0, p_a = 0.59$ during training (and $p_d = 0.31, p_a = 0.71$ during inference and certification). In Figure 4 (c) we use the same probabilities $p_d, p_a$ during training and inference.

In our experiments (Section 7), we also randomly sample different probabilities for training and inference to explore the joint parameter space of the training-time and inference-time smoothing parameters. That is, our search space is $[0, 1]^4$ when sampling different probabilities from $[0, 1]$ for the Pareto-plots in Figure 6 and Appendix H (we sample separately for training and inference).

# H   Detailed Results

We report certified accuracies in Figure 16 for the corresponding certified ratios in Figure 3. Moreover, we provide detailed results for the datasets Cora-ML, Citeseer, and PubMed. We show results for second-hop attacks against (1) smoothed GAT models in Figure 11, (2) smoothed GATv2 models in Figure 12, (3) smoothed GCN models in Figure 13, and (4) smoothed SMA models in Figure 14. We run 1,000 experiments for each combination, drawing random deletion and ablation probabilities from $[0, 1]$ for each experiment (sampling separately for training and inference). Lines connect dominating points on the Pareto front. Comparing results with and without skip-connection we observe that skip-connections allow higher node feature ablation probabilities while retaining high accuracy, which can yield better robustness-accuracy tradeoffs. Moreover, as discussed in Section 7, evaluating certificates in transductive settings comes with serious shortcomings. We nevertheless report such results in Figure 15 for a smoothed GAT model.

**Abstained predictions.** Our smoothed classifier abstains from predicting if $p_{v,y^*}(G) \leq \overline{p_{v,\tilde{y}}(G)}$. We show the ratio of abstained predictions for smoothed GAT models trained on Cora-ML in Figure 17 for different edge deletion probabilities $p_d$ and node feature ablation probabilities $p_a$. We use the same ablation probability during training and inference for this specific experiment. We observe that our smoothed classifier abstains for rather large probabilities. Future work could introduce novel architectures and training techniques to further diminish the effect of abstained predictions.

**Experiments on ogbn-arxiv.** We run additional experiments and compute certificates for the larger graph ogbn-arixv with 169,343 nodes, 128 attributes and 40 classes (Hu et al., 2020). We adopt their transductive setting, implement two-layer smoothed GCNs with skip-connection and compute certificates for 100 randomly chosen test nodes. In Figure 18 we show results for $p_d = 0.1, p_a = 0.4$ during training, and $p_d = 0.3, p_a$=0.8 during inference and certification. Notably, we can certify GNNs for such large graphs. However, our approach only achieves $53\%$ clean accuracy in this setting.

---

[5] `https://pytorch-geometric.readthedocs.io`

Future work could develop novel architectures and training procedures to improve clean accuracy under our smoothing distribution.

**Experiments with different confidence levels.** We conduct additional experiments with varying confidence levels $\alpha$ and Monte-Carlo samples. We observe strong guarantees for even smaller confidence levels, requiring little computational efforts. The underlying reason for this is that the theoretical largest certifiable radius of our certificates is bounded, only determined by the edge deletion probability $p_d$ and node feature ablation probability $p_a$, and therefore cannot increase by changing $\alpha$. Our certificates are thus less sensitive to changes in $\alpha$ compared to Neyman-Pearson-based certificates (Bojchevski et al., 2020).

In fact, the difference in certifiable robustness for $\alpha = 0.05$ and $\alpha = 0.0001$ is already extremely small when drawing just $2,000$ Monte-Carlo samples (Figure 19 a). We only observe differences in robustness for considerably small amounts of Monte-Carlo samples (Figure 19 b). Drawing 2,000 samples takes only 12 seconds on Cora-ML on average. This is significantly faster compared to all previous probabilistic certificates for GNNs that use up to $10^6$ Monte-Carlo samples (compare (Bojchevski et al., 2020)). In additional experiments, we also found that the classification accuracy is high for just a few thousand Monte-Carlo samples (Figure 20).

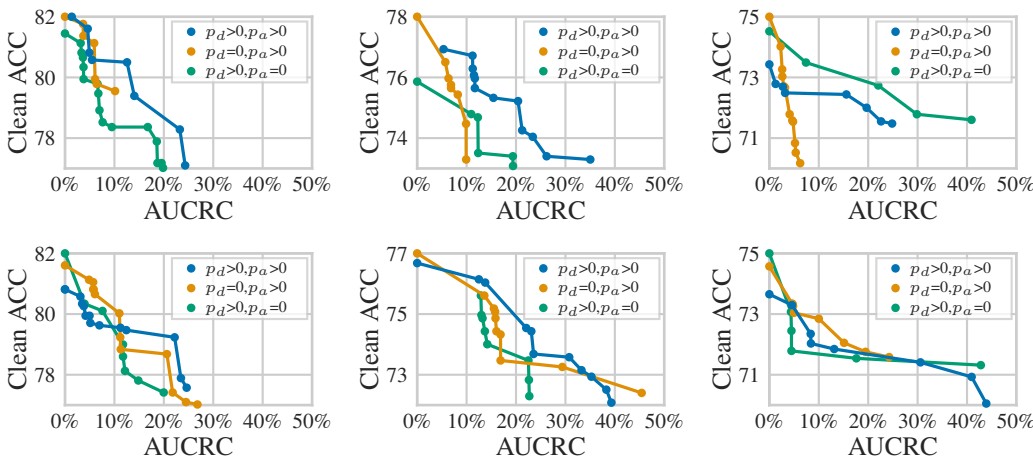

Figure 11: Robustness-accuracy tradeoffs for second-hop attacks against *smoothed GAT* on Cora-ML, Citeseer and PubMed (columns). Top row without skip-connection, bottom row with skip-connection. Lines connect dominating points on the Pareto front.

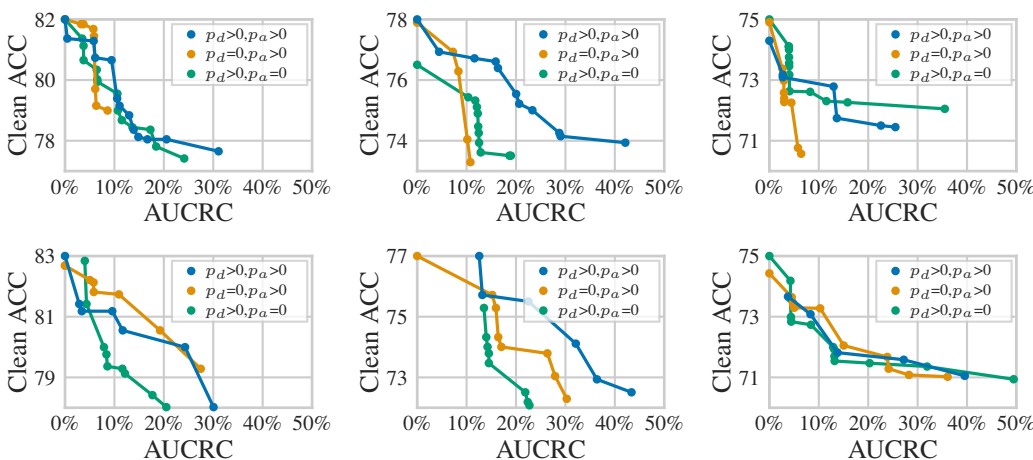

Figure 12: Robustness-accuracy tradeoffs for second-hop attacks against *smoothed GATv2* on Cora-ML, Citeseer and PubMed (columns). Top row without skip, bottom row with skip-connection.

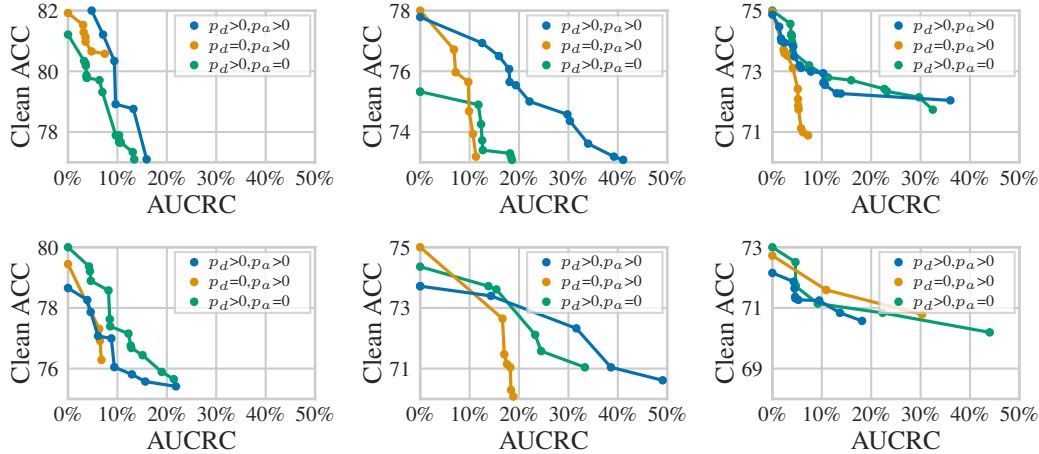

Figure 13: Robustness-accuracy tradeoffs for second-hop attacks against *smoothed GCN* on Cora-ML, Citeseer and PubMed (columns). Top row without skip-connection, bottom row with skip-connection.

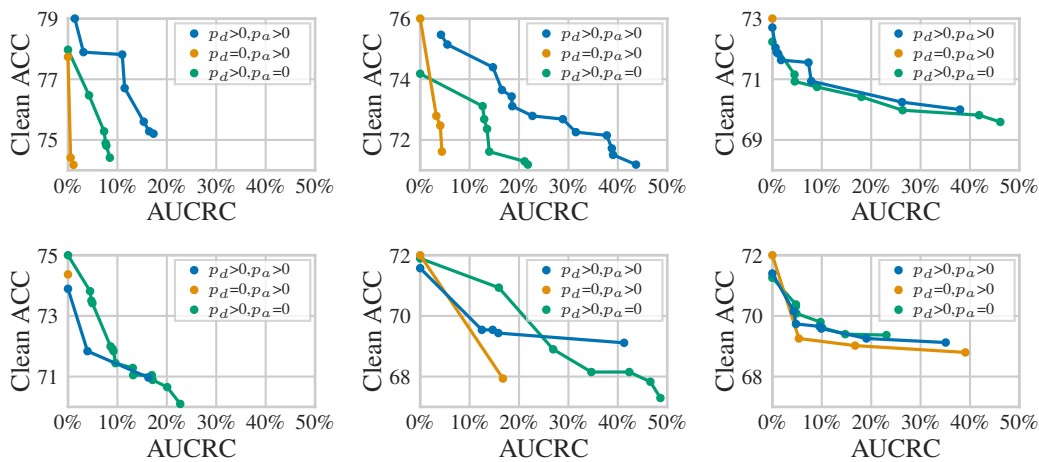

Figure 14: Robustness-accuracy tradeoffs for second-hop attacks against *smoothed SMA* on Cora-ML, Citeseer and PubMed (columns). Top row without skip-connection, bottom row with skip-connection.

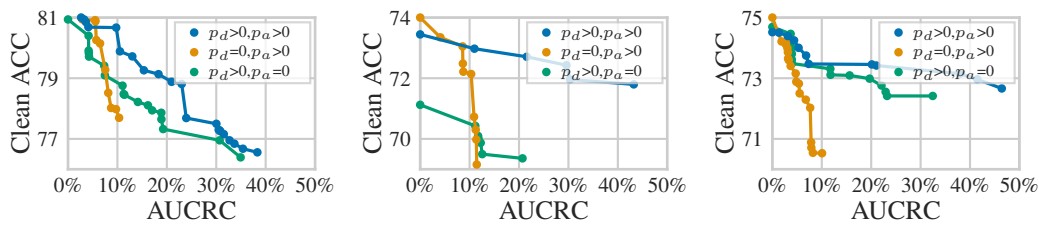

Figure 15: Transductive learning setting: Robustness-accuracy tradeoffs for second-hop attacks against *smoothed GAT* on Cora-ML, Citeseer and PubMed. Experiments without skip-connection.

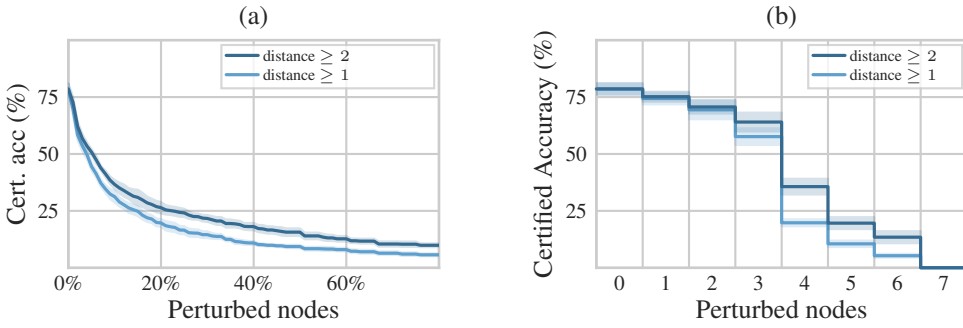

Figure 16: Certified accuracies for the setting of Figure 3 – Smoothed GAT on Cora-ML: (a) Robustness at different distances ($p_d$=0.31, $p_a$=0.794, with skip-connection, ACC=0.79).

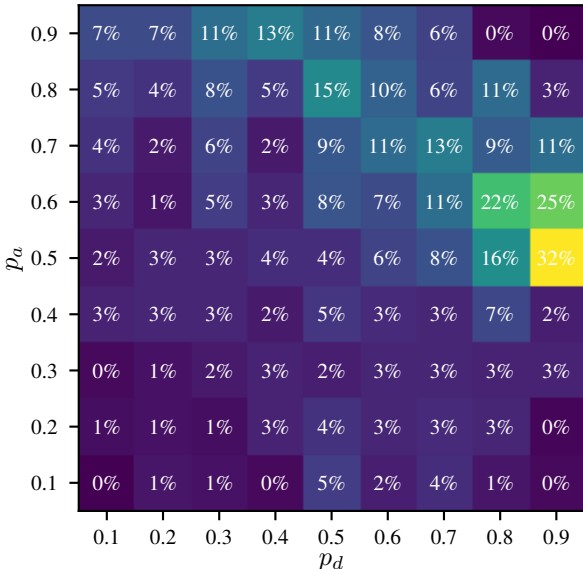

Figure 17: Abstained ratios of smoothed GAT models trained on Cora-ML for different edge deletion probabilities $p_d$ and node feature ablation probabilities $p_a$.

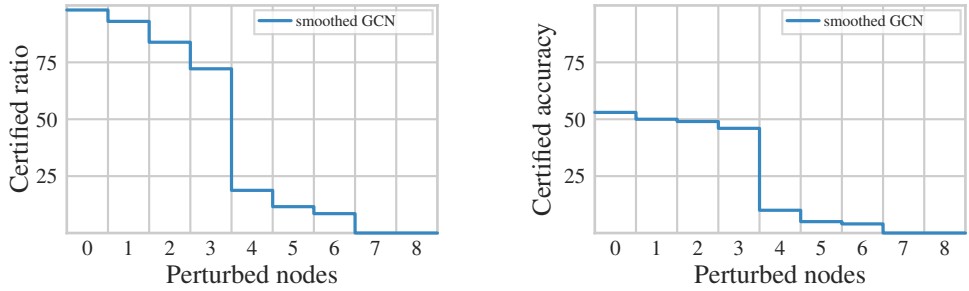

Figure 18: Certified ratio and accuracy for smoothed two-layer GCN on ogbn-arxiv. We certify 100 randomly selected test nodes in the graph. Certificates for nodes with distance 2 to the target node.

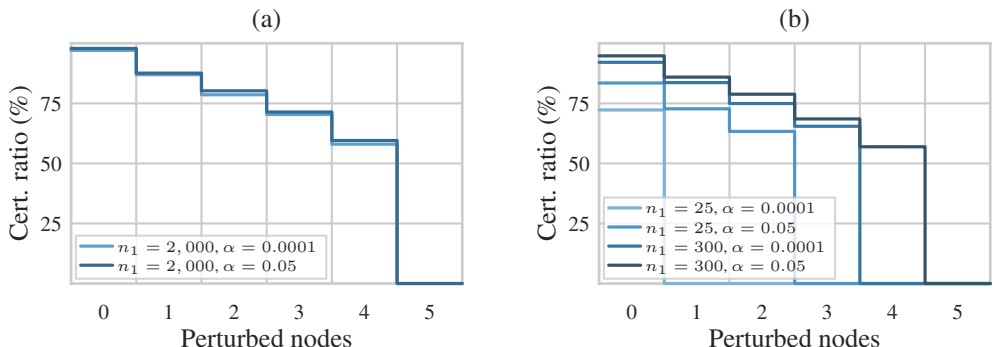

Figure 19: Certified ratio of smoothed GAT on Cora-ML ($p_a = 0.84$, $p_d = 0$, with skip-connection) for different confidence levels $\alpha$ and number of Monte-Carlo samples $n_1$. The difference in robustness is already considerably small for just 2,000 samples.

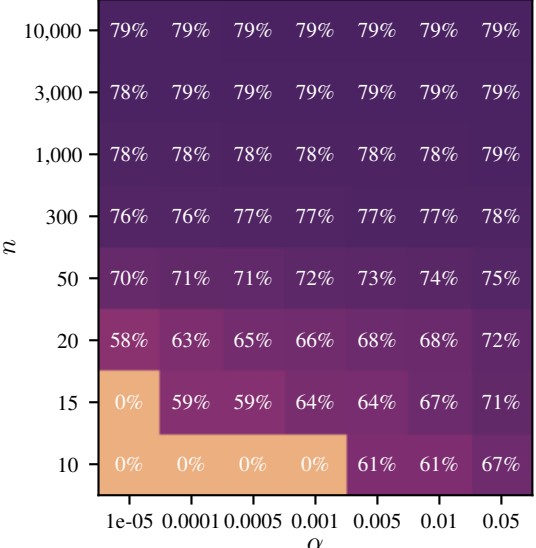

Figure 20: Clean accuracy of smoothed GAT on Cora-ML ($p_a = 0.84$, $p_d = 0$, with skip-connection). for varying number of confidence levels $\alpha$ and Monte-Carlo samples $n$. For $\alpha = 0.05$ the clean accuracy is high for just $1,000$ samples. For smaller $\alpha$, the certification accuracy decreases only slightly. Drawing more than $3,000$ samples is not necessary except for extremely small confidence levels such as $\alpha = 0.00001$.

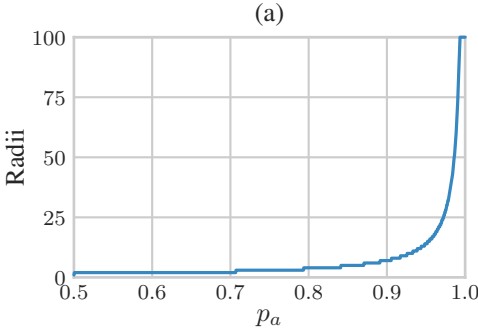 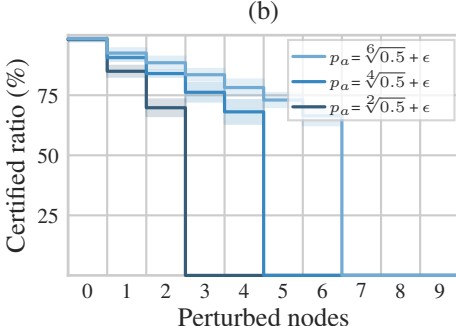

Figure 21: Visualizing Proposition 3. (a) Theoretically maximally certifiable radius for given node ablation probability $p_a$. (b) Certified ratio of smoothed GAT trained on CoraML for different node ablation probabilities ($p_d = 0$, $\epsilon = 0.01$). Note: $\sqrt[2]{0.5} \approx 0.71$, $\sqrt[4]{0.5} \approx 0.84$ and $\sqrt[6]{0.5} \approx 0.89$.

## I   On Neyman-Pearson and Ablation Certificates

There are currently two types of randomized smoothing certificates for discrete data: The certificates of Lee et al. (2019) and Bojchevski et al. (2020) are based on the Neyman-Pearson Lemma (Neyman and Pearson, 1933), and we therefore call them Neyman-Pearson-based certificates. The other certificates are ablation-based (Levine and Feizi, 2020b,a; Liu et al., 2021). We show that largest certifiable radius of ablation-based certificates is bounded indepdentent of the classifier, which is not the case for Neyman-Pearson-based certificates (see discussion in Section 6).

In ablation-based certificates, the bounding constant $\Delta$ determines the probability mass of the distribution $p_{v,y}(G)$ over labels $y$ that the worst-case adversary controls. This probability mass $\Delta$ is independent of the classifier $f$ and distribution $p_{v,y}(G)$ and solely determined by the smoothing distribution. Although the final certificates still depend on the classifier $f$, the largest certifiable radius of such ablation-based certificates is bounded as we show for our interception smoothing certificates:

Note again that $\Delta$ does not depend on the base GNN $f$: the probability to receive at least one message from a perturbed node is only characterized by the number of perturbed nodes $\rho$, and the probabilities $p_d$ for edge deletion and $p_a$ for node ablation. Moreover, $\Delta$ is monotonously increasing in $\rho$, since the probability to receive messages from perturb nodes increases the more nodes adversaries control. Interestingly, since $\Delta$ is monotonously increasing in $\rho$, there exists a largest certifiable radius that depends on the graph structure and changes for each target node (assuming fixed $p_d, p_a$). In the special case of node ablation smoothing, we can directly determine the largest certifiable radius:

**Proposition 3.** *Given fixed $p_a > 0$ and $p_d = 0$, it is impossible to certify a radius $\rho$ if $p_a \leq \sqrt[\rho]{0.5}$.*

*Proof.* Due to Corollary 3 and Corollary 1, we only get certificates if $\Delta < \frac{1}{2}$, i.e. the adversary should not control more than half of the distribution $p_{v,y}(G)$ over $y$. Thus:

$$\Delta < \frac{1}{2} \overset{(1)}{\Leftrightarrow} 1 - p_a^\rho < \frac{1}{2} \Leftrightarrow p_a^\rho > \frac{1}{2} \Leftrightarrow p_a > \sqrt[\rho]{0.5}$$

since the root is monotonously increasing and $p_a > 0$. Further, (1) stems from Proposition 2. Thus we need an ablation probability of at least larger than $\sqrt[\rho]{0.5}$ to certify a radius of $\rho$. $\qquad \square$

Proposition 3 allows us to directly determine the largest certifiable radius for given $p_a$. We visualize this largest radius for different ablation probabilities in Figure 21 (a). Theoretically, we can only certify large radii for relatively large ablation probabilities: For example, to theoretically certify a radius of 10, we already need an ablation probability of more than $\sqrt[10]{0.5} \approx 0.933$. Proposition 3 implies that we cannot certify any radius for ablation probabilities $p_a \leq 0.5$ (cf. Figure 2). Moreover, we can certify a radius of only 1 for ablation probabilities between $\sqrt[1]{0.5} = 0.5$ and $\sqrt[2]{0.5} \approx 0.707$. Note, however, that this is only a theoretical consideration and that the certificate also depends on the label probabilities $p_{v,y^*}(G)$ and $p_{v,\tilde{y}}(G)$ in practice (Figure 21 b), where we observe that the certified ratio drops to zero when the largest certifiable radius is passed.

# J Message-passing-aware Derandomization

As discussed in Section 6, our certificates are probabilistic and hold with a certain confidence level $\alpha$. Here we present alternative, deterministic certificates using a simplified smoothing distribution that just deletes nodes instead of ablating their features. We believe that future work can build upon it towards even more efficient and scalable derandomization schemes. Specifically, our derandomized certificates come with the following advantages: First, they are deterministic, exact certificates and hold independent of a confidence level. Second, the smoothed classifier never abstains from making a prediction (we resolve draws by whatever index comes first). Third, with more computation time we obtain more derandomized certificates. This is in continuation to probabilistic certificates that can be improved using more Monte-Carlo samples (Cohen et al., 2019).

**Simplified smoothing distribution.** We define a smoothed classifier that classifies node $v$ in $G$ as follows: Consider a retention constant $k \in \mathbb{N}$ that represents the number of nodes not deleted (retained) in the receptive field. Then the smoothed classifier $g$ predicts class $y$ with the largest probability $p_{v,y}(G)$ that $f$ classifies $v$ as $y$ under uniform deletion of all but $k$ nodes:

$$g_v(G) \triangleq \arg\max_y p_{v,y}(G) \qquad p_{v,y}(G) \triangleq p_{\mathcal{K} \sim \mathcal{U}(d,k)}(f(\mathcal{R}_{\mathcal{K}}) = y)$$

where $\mathcal{R}_{\mathcal{K}}$ encodes the deletion of all nodes in the receptive field of target node $v$ except those indexed by $\mathcal{K}$, and $f(\mathcal{R}_{\mathcal{K}})$ denotes the predicted class of $f$ for target node $v$ given ablated graph $\mathcal{R}_{\mathcal{K}}$ (omitting $v$ for conciseness). We further denote the indexing of nodes $\mathcal{K}$ as follows: Define the set of all $k$ unique indices in $[d] \triangleq \{1, \ldots, d\}$ including 0 as $B(d,k) = \{\{0\} \cup M : M \in \mathcal{P}([d]) \wedge |M| = k\}$, where $\mathcal{P}$ denotes the power set (w.l.o.g. we index target nodes as 0). For example, $\mathcal{K} = \{0, 1, 3, 6\} \in B(d,k)$ for retention constant $k = 3$ and receptive field size $d = 10$. Note that $|\mathcal{K}| = k + 1$ for $\mathcal{K} \in B(d,k)$ but $|B(d,k)| = \binom{d}{k}$ since we never delete the target node. Finally, let $\mathcal{U}(d,k)$ denote the uniform distribution over $B(d,k)$.

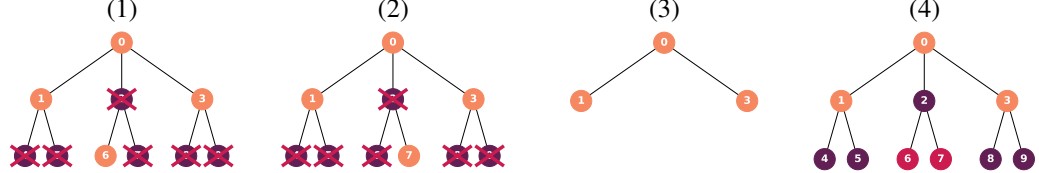

Figure 22: Given a receptive field with 10 nodes, target node 0 and $k = 3$. (1) If we keep nodes $\mathcal{K} = \{0, 1, 3, 6\}$ and delete all other nodes, node 6 is disconnected. (2) If we keep nodes $\mathcal{K} = \{0, 1, 3, 7\}$ and delete all other nodes, node 7 is disconnected. (3) In both cases, only the nodes $\mathcal{S}(\mathcal{K}) = \{0, 1, 3\}$ affect the prediction. (4) In the algorithm: Given $\mathcal{S} = \{0, 1, 3\}$ with neighborhood $\mathcal{N}_{\mathcal{S}} = \{2, 4, 5, 8, 9\}$. Choosing $k + 1 - |\mathcal{S}| = 1$ further nodes, we find that $\mathcal{S}$ is a reduced representative $\mathcal{S}(\mathcal{K})$ since there are $|V_v| - |\mathcal{N}_{\mathcal{S}}| - |\mathcal{S}| = 10 - 5 - 3 = 2$ nodes to choose from (6 and 7).

Computing $p_{v,y^*}(G)$ and $p_{v,\tilde{y}}(G)$ exactly is challenging. One naive approach would be to simply iterate over the support of the smoothing distribution (all possible node deletions). For small receptive fields, the number of possible combinations to sample $k$ out of $d$ nodes may be small, allowing us to enumerate all possibilities. However, this may be infeasible for larger receptive fields. Still, similar to how we use the message-passing structure for certification, we can also leverage it here to partition the support of the simplified smoothing distribution into a smaller number of equivalence classes.

Specifically, we observe: First, when uniformly deleting nodes in the receptive field, some of the remaining nodes $\mathcal{K}$ may be disconnected from the target node. Moreover, disconnected nodes will not affect the prediction for the target node. Second, several possibilities for $\mathcal{K}$ may share the same nodes that are still connected to $v$ (see examples in Figure 22). This means that different possibilities for $\mathcal{K}$ will lead to the same prediction by $f$, but the full enumeration of all possibilities is suboptimal: We wish to avoid redundant evaluations since the evaluation of the base classifier $f$ may be costly.

We observe that the connectivity explained above induces an equivalence relation: All sampled nodes $\mathcal{K}$ that share the same nodes connected to $v$ can be grouped into equivalence classes $[\mathcal{K}]$. For any representative $\mathcal{K}$ of $[\mathcal{K}]$ we denote the nodes still connected to $v$ as $\mathcal{S}(\mathcal{K})$. We call $\mathcal{S}(\mathcal{K})$ a reduced representative, since it represents a reduced form of $\mathcal{K}$ and only contains the nodes from which the target node will receive messages. Note that $\mathcal{S}(\mathcal{K})$ is unique for all representatives $\mathcal{K}$.

Formally, given receptive field $\mathcal{R}$ with $d+1$ nodes and index $\mathcal{K} \in B(d,k)$ of $k+1$ nodes. Consider the subgraph $\mathcal{R}_\mathcal{K}$ induced by $\mathcal{K}$. We observe that not necessarily all nodes in $\mathcal{R}_\mathcal{K}$ have to be connected to the target node. Thus, different $\mathcal{K} \in B(d,k)$ will result in same prediction of the base classifier. Let $\mathcal{S}(\mathcal{K}) \subseteq \mathcal{K}$ denote all nodes indexed by $\mathcal{K}$ without the disconnected nodes. Put differently, $\mathcal{S}(\mathcal{K})$ stands for nodes still connected to the target node (see example in Figure 22). Then:

**Proposition 9.** *The definition of $\mathcal{S}(\mathcal{K})$ induces an equivalence relation $\sim$ over $B(d,k)$ given by $\mathcal{K} \sim \mathcal{K}' \Leftrightarrow \mathcal{S}(\mathcal{K}) = \mathcal{S}(\mathcal{K}')$ and eq. classes $[\mathcal{K}] := \{\mathcal{K}' \in B(d,k) : \mathcal{K} \sim \mathcal{K}'\}$ for $\mathcal{K} \in B(d,k)$.*

*Proof.* Reflexivity, symmetry and transitivity hold by the definition of sets. $\qquad\square$

The equivalence relation $\sim$ partitions $B(d,k)$ into disjoint equivalence classes, denoted by the quotient set $B(d,k)/\sim \;\triangleq\; \{[\mathcal{K}] \mid \mathcal{K} \in B(d,k)\}$. The set $\mathcal{S}(\mathcal{K})$ is uniquely defined for each equivalence class $[\mathcal{K}]$ in $B(d,k)/\sim$. We therefore call $\mathcal{S}(\mathcal{K})$ with $1 \le |\mathcal{S}(\mathcal{K})| \le k+1$ the **reduced representative** of $[\mathcal{K}]$. Note that we have $|\mathcal{S}(\mathcal{K})| = k+1 \Leftrightarrow \mathcal{S}(\mathcal{K}) = \mathcal{K}$ and $|[\mathcal{K}]| = 1$. We further call $\mathbb{S} = \{\mathcal{S}(\mathcal{K}) \mid \mathcal{K} \in B(d,k)\}$ the **complete set of reduced representatives**. Note that $\mathbb{S} \cong B(d,k)/\sim$ and thus $|\mathbb{S}| = |B(d,k)/\sim|$.

To efficiently derandomize our certificates, we can leverage the fact that we only need a complete set of reduced representatives $\mathbb{S}$ to compute the label probabilities $p_{v,y}(G)$. Given $\mathbb{S}$, we only have to evaluate $f$ *once* for each reduced representative $\mathcal{S}(\mathcal{K}) \in \mathbb{S}$:

**Corollary 5.** *Given the complete set of reduced representatives $\mathbb{S}$, the label probabilities are:*

$$p_{v,y}(G) = \binom{d}{k}^{-1} \sum_{\mathcal{S} \in \mathbb{S}} \mathbb{I}[f(\mathcal{R}_\mathcal{S}) = y] \cdot \beta_\mathcal{S}$$

*where $\mathbb{I}[f(\mathcal{R}_\mathcal{S}) = c]$ indicates whether $f$ classifies the target node $v$ in subgraph $\mathcal{R}_\mathcal{S}$ as class $c$, and $\beta_\mathcal{S}$ is the size of an equivalence class, $\beta_\mathcal{S} = |[\mathcal{K}]|$. We write $\mathcal{S} \triangleq \mathcal{S}(\mathcal{K})$ and omit $v$ for conciseness.*

*Proof.* For all $\mathcal{K}, \mathcal{K}' \in B(d,k)$ with $\mathcal{K} \sim \mathcal{K}'$ we have $f_v(\mathcal{R}_\mathcal{K}^v) = f_v(\mathcal{R}_{\mathcal{K}'}^v) = f_v(\mathcal{R}_{\mathcal{S}(\mathcal{K})}^v)$ as only information from nodes of the reduced representative $\mathcal{S}(\mathcal{K})$ can be passed to the target node (other nodes are disconnected). Thus, instead of evaluating $f_v(\mathcal{R}_\mathcal{K}^v(G))$ for all $\mathcal{K} \in B(d,k)$ we only have to evaluate $f_v(\mathcal{R}_{\mathcal{S}(\mathcal{K})}^v(G))$ for each $\mathcal{S}(\mathcal{K}) \in \mathbb{S}$. To do so we have to count $f_v(\mathcal{R}_{\mathcal{S}(\mathcal{K})}^v(G)) = i$ exactly $\beta_\mathcal{S} = |[\mathcal{K}]|$ times. Further, as we uniformly sample $\mathcal{K}$ from $\mathcal{U}(d,k)$ over $B(d,k)$, we have to scale the possibilities by $|B(d,k)|^{-1}$, which corresponds to the inverse binomial coefficient above. $\quad\square$

Hence, we can compute the label probabilities $p_{v,y}(G)$ exactly for larger receptive fields if the number of equivalence classes $|\mathbb{S}|$ is small and we have an efficient algorithm to compute $\mathbb{S}$ and $\beta_\mathcal{S}$. We propose such algorithm by exploiting the sparsity of graphs as follows:

We successively enumerate all possible connected subgraphs of the receptive field $\mathcal{R}$ indexed by $\mathcal{S}$ that contain the target node and at most $k$ further nodes. Let $\mathcal{S}$ denote indices of such subgraph of $\mathcal{R}$ and $\mathcal{N}_\mathcal{S}$ the neighborhood of $\mathcal{S}$ in $\mathcal{R}$. If $\mathcal{S}$ contains $k+1$ nodes, then all $k+1$ nodes will be connected to the target node and $\mathcal{S}$ is already a representative with $\beta_\mathcal{S} = 1$. If $\mathcal{S}$ contains less than $k+1$ nodes, then $\mathcal{S}$ corresponds to a reduced representative if we can choose the remaining $k+1-|\mathcal{S}|$ nodes such that they are disconnected. Therefore, the main idea of our algorithm is that the size $\beta_\mathcal{S}$ is just a binomial coefficient: The number of disconnected nodes is given by $|V_v| - |\mathcal{N}_\mathcal{S}| - |\mathcal{S}|$, out of which we have to choose $k+1-|\mathcal{S}|$ nodes to augment $\mathcal{S}$ to set of $k+1$ nodes (where $V_v$ denote nodes in the receptive field):

$$\beta_\mathcal{S} = \binom{|V_v| - |\mathcal{N}_\mathcal{S}| - |\mathcal{S}|}{k+1-|\mathcal{S}|}$$

If $\beta_\mathcal{S} > 0$, there must exist a representative $\mathcal{K}$ such that the reduced representative $\mathcal{S}(K)$ corresponds to $\mathcal{S}$, that is $\mathcal{S} = \mathcal{S}(\mathcal{K})$ (compare (4) in Figure 22 for an example). Finally, our algorithm enumerates all possible $\mathcal{S}$ by recursively augmenting $\mathcal{S}$ with nodes from the neighborhood of $\mathcal{S}$ (compare algorithm 1). This way, we exploit the sparsity of graphs to find all reduced representatives $\mathbb{S}$ that avoid disconnected nodes.

**Algorithm 1:** Compute complete set of reduced representatives $\mathbb{S}$ and equivalence class sizes $\beta_\mathcal{S}$

---

**Input:** Index 0 of target node $v$, Receptive field $\mathcal{R}^v = (V_v, E_v)$, Retention constant $k$
$\mathcal{S} \leftarrow \{0\}$
**Output:** EQCGeneration($\mathcal{S}$, $V_v$, $E_v$, $k$)

**Function** EQCGeneration($\mathcal{S}$, $V_v$, $E_v$, $k$):

> $R \leftarrow \{\}$
> **if** $|\mathcal{S}| = k + 1$ **then**
> > **return** $\{(\mathcal{S}, 1)\}$
>
> **end**
> $\mathcal{N}_\mathcal{S} \leftarrow \{w \in V_v \setminus \mathcal{S} \mid \exists u \in \mathcal{S} : (w, u) \in E_v\}$         // $\mathcal{O}(|V_v|)$
> $\beta_\mathcal{S} \leftarrow$ binom$(|V_v| - |\mathcal{N}_\mathcal{S}| - |\mathcal{S}|, k + 1 - |\mathcal{S}|)$
> **if** $\beta_\mathcal{S} > 0$ **then**
> > $R \leftarrow \{(\mathcal{S}, \beta_\mathcal{S})\}$
>
> **end**
> **for** $w \in \mathcal{N}_\mathcal{S}$ **do**                                       // $\mathcal{O}(|V_v|)$
> > $R \leftarrow R \cup$ EQCGeneration($\mathcal{S} \cup \{w\}$, $V_v$, $E_v$, $k$)
>
> **end**
> **return** $R$

---

Note that in algorithm 1, $V_v$ denotes nodes in the receptive field of classifier $f$ with respect to target node $v$, and $E_v$ the edges in the receptive field.

**Lemma 2** (Correctness of algorithm 1). *Let $\mathcal{S}$ with $0 \in \mathcal{S} \subseteq V_v$ be a set of at most $k + 1$ nodes $1 \leq |\mathcal{S}| \leq k + 1$ such that all nodes indexed by $\mathcal{S}$ are connected to the target node in $\mathcal{R}$. We denote the neighbors of $\mathcal{S}$ in $\mathcal{R}$ as $\mathcal{N}_\mathcal{S} \triangleq \{w \in V_v \setminus \mathcal{S} \mid \exists u \in \mathcal{S} : (w, u) \in E_v\}$. When we define the following binomial coefficient as*

$$\beta_\mathcal{S} \triangleq \binom{|V_v| - |\mathcal{N}_\mathcal{S}| - |\mathcal{S}|}{k + 1 - |\mathcal{S}|} \in \mathbb{N}.$$

*then there exists a representative $\mathcal{K} \in B(d, k)$ such that $\mathcal{S}$ is a reduced representative for the equivalence class $[\mathcal{K}]$ if $\beta_\mathcal{S} > 0$. Then we have $\beta_\mathcal{S} = |[\mathcal{K}]|$.*

*Proof.* First note that for a given set $\mathcal{S}$ as defined above we can partition $V_v$ into three disjoint sets $V_v = \mathcal{S} \uplus \mathcal{N}_\mathcal{S} \uplus \mathcal{N}_r$ with $\mathcal{S}$ and $\mathcal{N}_\mathcal{S}$ defined as above, and the disconnected nodes $\mathcal{N}_r \triangleq V_v \setminus (\mathcal{S} \cup \mathcal{N}_\mathcal{S})$. We thus have $|\mathcal{N}_r| = |V_v| - |\mathcal{N}_\mathcal{S}| - |\mathcal{S}|$. Now we distinguish the following cases:

Case 1: $|\mathcal{S}| = k + 1$

We have $|V_v| - |\mathcal{N}_\mathcal{S}| - |\mathcal{S}| \in \mathbb{N}_0$ and $\beta_\mathcal{S} = 1 > 0$. Thus for $|\mathcal{S}| = k + 1$ the condition is trivially fulfilled and we have that $\mathcal{K} \triangleq \mathcal{S}$ is already a representative with $|[\mathcal{K}]| = 1$ as discussed before. Note that this does not mean that all sets with $k + 1$ nodes are representatives, as we still have the connectivity constraint for nodes in $\mathcal{S}$.

Case 2: $|\mathcal{S}| < k + 1$

We have $\beta_\mathcal{S} > 0 \Leftrightarrow |V_v| - |\mathcal{N}_\mathcal{S}| - |\mathcal{S}| \geq k + 1 - |\mathcal{S}| \Leftrightarrow |\mathcal{N}_r| \geq k + 1 - |\mathcal{S}|$ where the latter means that we can choose the remaining $k + 1 - |\mathcal{S}|$ nodes from $\mathcal{N}_r$ to augment $\mathcal{S}$ to representative $\mathcal{K}$ of the equivalence class $[\mathcal{K}]$ since then $|\mathcal{K}| = |\mathcal{S}| + k + 1 - |\mathcal{S}| = k + 1$. The corresponding size $|[\mathcal{K}]|$ is given by $\beta_\mathcal{S}$. $\qquad\qquad\square$

Finally, note that the equivalence classes and the algorithm are independent of the classifier $f$.

**Discussion.** In the worst case, we have $|\mathbb{S}| = |B(d,k)| = \binom{d}{k}$, but we enumerate $\sum_{i=0}^{k} \binom{d}{i} \geq \binom{d}{k}$ possibilities, as there are $\sum_{i=0}^{k} \binom{d}{i}$ candidates for reduced representatives in a fully connected graph. Therefore, in the worst case of fully connected graphs, directly enumerating all $\binom{d}{k}$ possibilities would be faster. In practice, however, we rather observe sparse graphs with $|\mathbb{S}| \ll |B(d,k)|$. The more sparse the receptive field, the less equivalence classes exist and the larger each equivalence class. Thus we exploit the sparsity of graphs to efficiently compute $\mathbb{S}$ and the corresponding sizes $|[\mathcal{K}]|$ for all equivalence classes $[\mathcal{K}]$.

Moreover, as our algorithm recursively enumerates all possible pairs $(\mathcal{S}, \beta_{\mathcal{S}})$, we can determine a stopping criterion at which we back off to Monte-Carlo sampling for estimating the label probabilities. To this end, if $R$ denotes the current set of $(\mathcal{S}, \beta_{\mathcal{S}})$ pairs with $\beta_{\mathcal{S}} > 0$, we know that $|R|$ is a lower bound on the number of equivalence classes, $|R| \leq |\mathbb{S}|$. By summing up $\beta_{\mathcal{S}}$ for all $(\mathcal{S}, \beta_{\mathcal{S}}) \in R$ we can determine the percentage of $|B(d,k)|$ that we already cover with $R$:

$$\sum_{(\mathcal{S}, \beta_{\mathcal{S}}) \in R} \beta_{\mathcal{S}} \leq \sum_{\mathcal{S}(\mathcal{K}) \in \mathbb{S}} |[\mathcal{K}]| = \binom{d}{k} = |B(d,k)|$$

This allows us to use the condition $\sum_{(\mathcal{S}, \beta_{\mathcal{S}}) \in R} \beta_{\mathcal{S}} > \tau'$ with threshold $\tau' \in \mathbb{N}$ as a stopping criterion. Using thresholds this way, our algorithm will always find more solutions in $\mathbb{S}$ given more time via larger thresholds. Note that we use $\binom{d}{k} > \tau$ in practice, since the binomial coefficient provides a fast upper bound for the number of equivalence classes $|\mathbb{S}|$.

### J.1 Evaluating Message-passing-aware Derandomization

Table 1: Smoothed classifier results for GCN trained on Cora-ML for different relative retention constants. Der.: Ratio of nodes with derandomized certificates. Eq.: Mean of unique receptive fields over all derandomized certificates. Acc.: Clean accuracy.

| | GCN on Cora-ML | | | | GCN on Citeseer | | | | GCN on PubMed | | | |
|---|---|---|---|---|---|---|---|---|---|---|---|---|
| $k_{rel}$ | Der. | Eq. | Abstained | Acc. | Der. | Eq. | Abstained | Acc. | Der. | Eq. | Abstained | Acc. |
| 0.01 | 0.87 | 0.22 | 6.27e-04 | 0.73 | 1.00 | 0.41 | 0.00e+00 | 0.65 | 0.94 | 0.15 | 0.00e+00 | 0.73 |
| 0.03 | 0.72 | 0.23 | 5.69e-04 | 0.73 | 0.94 | 0.42 | 0.00e+00 | 0.66 | 0.81 | 0.16 | 1.56e-03 | 0.73 |
| 0.10 | 0.50 | 0.28 | 5.02e-03 | 0.74 | 0.87 | 0.42 | 1.63e-03 | 0.65 | 0.61 | 0.19 | 4.24e-03 | 0.74 |
| 0.30 | 0.31 | 0.46 | 1.42e-02 | 0.80 | 0.73 | 0.53 | 7.61e-03 | 0.68 | 0.37 | 0.38 | 6.23e-03 | 0.77 |

**Relative retention constant.** Consider a small retention constant $k = 1$ for a node $v$ with $deg(v) < d_v - deg(v)$, where $d_v$ denotes the receptive field size (excluding the target node). Then the probability for selecting a direct neighbor of $v$ is low and the prediction of the smoothed classifier is merely based on the target node $v$ itself, which amounts to traditional i.i.d. prediction. Thus, for non-trivial robustness guarantees we use retention constants $k$ that are relative to the receptive field size: Given a fixed relative retention constant $k_{rel} \in [0,1]$, our smoothed classifier keeps $k = \lceil d_v \cdot k_{rel} \rceil \in \mathbb{N}$ nodes in the receptive field $\mathcal{R}$.[6] The ceiling operation ensures that we keep at least one additional node.

**Derandomization results.** Our certificates are deterministic for small receptive fields, and probabilistic for large receptive fields: we derandomize certificates if $\binom{d}{k}$ is smaller than a threshold $\tau$. If the number of possibilities to choose $k$ out of $d$ nodes is small, we can enumerate all possibilities and use $f$ to predict the class of $v$ for all possibilities. In our experiments we set $\tau = 100{,}000$. There are more possibilities to sample $k$ out of $d$ nodes for larger $k_{rel}$ and thus the ratio of deterministic certificates decreases (compare Table 1). For example, we can derandomize around 50% of the certificates for Cora-ML given $k_{rel} = 0.1$. We further derandomize more certificates for Citeseer than for Cora-ML, which can be explained by the fact that two-layer GNNs have larger receptive fields on Cora-ML. Note that the average degree in Cora-ML is 6, in Citeseer 3 and PubMed 4. Due to the derandomization we also hardly observe that the smoothed classifier abstains.

As discussed above, we avoid evaluating the base classifier $f$ for equivalent receptive fields. To represent the computations we avoid on average, we compute the mean of unique receptive fields $|\mathbb{S}|/|B(d,k)|$ for all derandomized certificates. For example, out of all derandomized certificates for $k_{rel} = 0.1$ on Cora-ML, we only have to evaluate 28% of all possibilities on average.

---

[6] As a disadvantage of this method, we have to process all receptive fields separately.