# OpenReview forum: "Randomized Message-Interception Smoothing: Gray-box Certificates for Graph Neural Networks"
_NeurIPS.cc/2022/Conference — NeurIPS 2022 Accept_

### Official Review · Reviewer_EH8E · 2022-06-29

**Rating:** 6
**Confidence:** 4
**Soundness:** 4 excellent
**Presentation:** 3 good
**Contribution:** 2 fair

**Summary:**

This paper proposes a robustness certification against adversarial attacks on graph data. The model will randomly ablate some nodes and delete some edges so that the adversarial impact on certain nodes will not propagate to the target node with high probability. They show that the pipeline achieves good robustness, especially at larger distances.

**Questions:**

In the experimental setup, how is the subgraphs generated exactly? Since the nodes are randomly sampled, does the edge only exist if both nodes are sampled? How do you guarantee that the graph is still connected and not too sparse?

I am curious why the authors keep mentioning gray-box certification in contrast to black-box. Since the certification is performed by the model itself, the model will for sure know the exact data as well as other information. So I can only imagine the white-box (which also satisfies gray-box and black-box) setting of the certification.

**Limitations:**

As mentioned in the weaknesses, I think the authors can improve by considering the edge-based attacks.

**Strengths And Weaknesses:**

## Strengths

* The robustness certification can be verified against arbitrary perturbation on the adversarial nodes. In the pipeline, the certification is achieved by blocking the messages from the adversarial nodes. Therefore, it can provide certification without restricting the perturbation magnitude on each node. This seems a realistic setting to me - in practical graph data settings, considering the number of adversarial nodes (users) is more important than considering the magnitude of adversarial perturbation on each node.

* The experimental evaluation is comprehensive. The authors evaluate their approach in different settings, e.g. with skip-connections and under different sparsifications. These approaches show the flexibility of their approach to different cases.


## Weaknesses:

* Only feature-based attack robustness is considered. Actually, a wide range of attacks on GNNs will focus on edge-modification attacks (e.g. [a,b]). The paper does not discuss the applicability of their approach against this type of attack. Actually, it feels to me that the proposed approach can still be adapted as long as both nodes of the edge are considered as adv nodes, although such a bound seems like a loose bound.

[a] Zügner D, Günnemann S. Adversarial Attacks on Graph Neural Networks via Meta Learning[C]//International Conference on Learning Representations. 2018.

[b] Xu K, Chen H, Liu S, et al. Topology attack and defense for graph neural networks: an optimization perspective[C]//Proceedings of the 28th International Joint Conference on Artificial Intelligence. 2019: 3961-3967.

* The certification result is node-specific and not applicable in practice. If I understand correctly, the exact certification provided by the algorithm is "given a target node, the adversarial perturbation on a specific node set can/cannot be certified to be robust", but not "given a target node, we can allow arbitrary N nodes to be attacked". The latter is a more useful certificate that can be achieved by previous approaches, while the current one is limited to a specified node set.

---

> ### Author Response · Authors · 2022-08-02
> **Response to Reviewer EH8E**
>
> Thank you for your review!
>
> ### Concerning our focus on feature perturbations (Comment 1)
> Thank you for pointing out that our method can be technically extended to certify the robustness against edge-modification attacks. In this paper, we focus on the novel problem of arbitrary feature manipulations of entire nodes since (1) there are already certificates against edge-modification attacks in the literature (see e.g. [1]), and (2) such feature-based attacks are becoming significantly stronger in recent years and are realistic scenarios [2,3].
>
> ### Concerning the choice of adversarial nodes (Comment 2)
> As you correctly pointed out, the threat model "given a target node, we can allow arbitrary N nodes to be attacked" may be a more common setting in practice. In fact, we discuss both scenarios in our work: given a specific set of adversarial nodes, or given N arbitrary adversarial nodes (compare Lines 137-139). We agree with you that the latter certificate may be more useful in general, and we therefore present a thorough evaluation of such certificates in Section 6.
>
> ### How are the subgraphs generated?
> While edges are randomly deleted, nodes are only randomly ablated (i.e. we replace their features with a special token). In practice, we sample edges to delete and nodes to ablate independently. Given we did not sample an edge for deletion, it remains in the graph even if its incident nodes are ablated since we do not delete nodes but only ablate their features.
>
> ### How do we guarantee that the graph is still connected and not too sparse?
> We do not guarantee that the graph remains connected, which is not required for the base models or for computing certificates. We control sparsity of the resulting graphs by adjusting the edge deletion probability p_d. With larger p_d, the sampled graphs become sparser and our certificates become stronger. In our extensive experiments with varying edge deletion probabilities (Section 6), we analyze the tradeoffs between accuracy and robustness of the models under randomized edge deletion (Figure 6).
>
> ### Why is our certificate a "gray-box" certificate?
> The term "gray-box" is not referring to the information the model has about its input data, but to the information the certificate has about the model. Existing certificates differ in their knowledge about the model (Section 7):
>
> For example, the certificate proposed in [4] considers the specific form of message-passing used in Graph Convolutional Networks and cannot be applied to other GNNs. Their certificate is a white-box certificate since it has full knowledge about the model.
>
> Another example is the method proposed in [1], which can certify any GNN model, even any function $f: \\{0, 1\\}^D \rightarrow \mathbb{R}^C$, since it does not take any properties of the function $f$ into account (like the fact that it is a message-passing GNN). Their certificate is a black-box certificate since it has no knowledge about the model.
>
> In contrast, our paper represents a novel contribution to combine the best of both worlds: our certificates are model-agnostic *and* consider the underlying message-passing structures of GNNs. As a result, we obtain significantly stronger robustness guarantees compared to existing certificates.
>
> ### References
> [1] Aleksandar Bojchevski, Johannes Gasteiger, Stephan Günnemann. Efficient Robustness Certificates for Discrete Data: Sparsity-Aware Randomized Smoothing for Graphs, Images and More. ICML 2020.
>
> [2] Jiaqi Ma, Shuangrui Ding, Qiaozhu Mei. Towards More Practical Adversarial Attacks on Graph Neural Networks. NeurIPS 2020.
>
> [3] Xu Zou, Qinkai Zheng, Yuxiao Dong, Xinyu Guan, Evgeny Kharlamov, Jialiang Lu, Jie Tang. TDGIA: Effective Injection Attacks on Graph Neural Networks. KDD 2021.
>
> [4] Daniel Zügner, Stephan Günnemann. Certifiable Robustness and Robust Training for Graph Convolutional Networks. KDD 2019.

---

> > ### Comment · Reviewer_EH8E · 2022-08-09
> > **Thank you for the explanation**
> >
> > Thank you for the detailed explanations from the author. My questions are solved and I would raise my score.

---

### Official Review · Reviewer_51To · 2022-07-13

**Rating:** 8
**Confidence:** 3
**Soundness:** 4 excellent
**Presentation:** 4 excellent
**Contribution:** 4 excellent

**Summary:**

This paper presents a technique to certify the robustness of Graph Neural Networks. Inspired by randomized smoothing, the proposed method takes advantage of the graph structure to get obtain various certificates (e.g., on accuracy). For the studied threat model, these certificates are tighter than proposed in the literature so far. The threat model consist of allowing an adversary to arbitrarily manipulate the features (i.e., emitted messages) of a subset of nodes.

**Questions:**

The paper is well-written. I would appreciate if the authors address the weaknesses stated above.

**Limitations:**

Yes, the limitations are reasonably addressed.

**Strengths And Weaknesses:**

Strengths:
* The authors are the first to truly take advantage of network structure to certify GNN (at least from a smoothing perspective).
* The results clearly demonstrate that they can certify a larger proportion of cases than prior methods (when the number of features increases).
* The threat model is fairly broad and could have practical applications

Weaknesses:
* It would help the reader to explain a bit why the studied threat model is interesting and when it arises in the real-world. The certification method which relies on smoothing can only be used for offline analysis or, if used online (let's say on the network of machine), will be prohibitly expensive to run (i.e., sampling of various network configurations).
* My knowledge of graph robustness is fairly limited, but I assume there has been a number of works (not specifically designed for machine learning applications) that also provide robustness to the threat model considered (e.g., r-robustness). Maybe expanding the literature survey slightly and expanding the experiments to demonstrate that r-robust networks are 100% certifiable (using the proposed approach) would help make a stronger and broader point.

---

> ### Author Response · Authors · 2022-08-02
> **Response to Reviewer 51To**
>
> Thank you for your review!
>
> ### Concerning real-world examples for our threat model (Comment 1)
> Please consider that we motivate our threat model in the introduction where we also provide a real-world example (Lines 22-24). Specifically in social networks, adversaries typically have full access to features of a few nodes in the graph. Further real-world examples include (1) vandalism against entities in public knowledge graphs [1], (2) fraud in online reviews [2], and (3) many security-related applications in financial or medical domains [3] where adversaries can also control entire nodes in the graph. Beyond that, our certificates also provide helpful information on the robustness of GNNs under non-adversarial perturbations including incomplete data, random perturbations, noisy signals, and data measuring errors. We included these additional examples and justifications for the threat model in Appendix L, and we will consider them for the introduction of the camera-ready version.
>
> ### Concerning computational cost (Comment 1)
> Our certificates are more sample-efficient than existing probabilistic certificates for GNNs. As we discuss in Lines 263-273, our method is significantly more efficient than existing work since 2,000 samples are sufficient to obtain strong guarantees, which takes only a few seconds in practice. We believe future work could even improve upon our results towards more robust models in online settings. Moreover, an offline analysis is also useful for the defender since e.g. knowing how robust a model is can help defenders to decide whether the model is ready to run in production.
>
> ### Concerning r-robustness (Comment 2)
> Thank you for pointing out this interesting connection to the r-robustness literature. We are not familiar with this field so far and the short answer period is not enough time to carefully consider all connections between the two fields and potential implications. From an introductory read of the literature, r-robust graphs have the property that we can remove r-1 nodes from the neighborhood of every remaining node such that the resulting graph is still connected, and certain algorithms can still find consensus under r-robust graphs even if there are malicious nodes [4,5]. They have very different goals as r-robustness is about reachability, while we want to limit the chances of adversarial messages to reach target nodes. Moreover, it appears that high r-robustness implies low certifiable robustness since higher connectivity yields lower probability to intercept messages. But we will need to look deeper into the literature about r-robustness and consider the connection to consensus algorithms more carefully until the camera-ready deadline.
>
> ### References
> [1] Stefan Heindorf, Martin Potthast, Benno Stein, Gregor Engels: Vandalism Detection in Wikidata. CIKM 2016.
>
> [2] Bryan Hooi, Neil Shah, Alex Beutel, Stephan Günnemann, Leman Akoglu, Mohit Kumar, Disha Makhija, Christos Faloutsos. BIRDNEST: Bayesian Inference for Ratings-Fraud Detection. SDM 2016.
>
> [3] Leman Akoglu, Hanghang Tong, Danai Koutra: Graph based anomaly detection and description: a survey. Data Min. Knowl. Discov. 29(3): 626-688 (2015)
>
> [4] Haotian Zhang, Elaheh Fata, Shreyas Sundaram: A Notion of Robustness in Complex Networks. IEEE Trans. Control. Netw. Syst. 2(3): 310-320 (2015)
>
> [5] Heath LeBlanc, Haotian Zhang, Xenofon D. Koutsoukos, Shreyas Sundaram: Resilient Asymptotic Consensus in Robust Networks. IEEE J. Sel. Areas Commun. 31(4): 766-781 (2013)

---

> > ### Comment · Reviewer_51To · 2022-08-03
> > **Thank you for the reply**
> >
> > Thank you for the reply. My questions have been answered so far, I will keep monitoring the discussion with other reviewers.

---

### Official Review · Reviewer_dqSE · 2022-07-17

**Rating:** 6
**Confidence:** 2
**Soundness:** 3 good
**Presentation:** 4 excellent
**Contribution:** 3 good

**Summary:**

This paper presents a new gray-box certification method for graph neural networks (GNNs) using smoothing. Unlike previous approaches, it uses utilizes the underlying message passing based principle (e.g., deleting edges, ablating nodes, etc) to obtain a stronger guarantee. To achieve certification, the authors first derived the worst case change in label probability and then proposed practical way to give a sound lower bound.


**Questions:**

Can you show results with different confidence levels and see how strong the method is with a smaller alpha? How many samples are required and what is the certification accuracy for different levels of alpha?


**Limitations:**

Yes, the author discussed limitations of this work explicitly and I see no obvious negative societal impacts.

**Strengths And Weaknesses:**

Strength:

1. This method can give stronger certification compared to previous approach.
2. The theoretical results presented in this paper are novel and practical.
3. Experimental evaluation shows both speed and certification benefits compared to pure black box methods.

Weaknesses:

1. I am a bit concerned about the confidence level alpha=0.05 used for certification, because it means that the guaranteed robustness only holds with a probability of 95%. Some existing works such as (Zugner and Gunnemann 2019)  using deterministic certification methods do not have this limitation.

---

> ### Author Response · Authors · 2022-08-02
> **Response to Reviewer dqSE**
>
> Thank you for your review!
>
> We choose $\alpha=0.05$ since this is a standard value used in the literature (see e.g. [1]). Nonetheless, we share your concerns and therefore conducted additional experiments using smaller confidence levels (see added Appendix M). Notably, our method still yields strong guarantees for significantly smaller confidence levels such as alpha=0.0001, for which our robustness guarantees hold with probability 99,99%. We believe that we can present all results for smaller confidence levels in the camera-ready version. Please also consider our additional elaboration in Appendix M where we explain that alpha has just a minor effect on the strength of our certificates.
>
> ### How strong is the method with smaller alpha and how many Monte-Carlo samples are required?
> We obtain strong guarantees for smaller confidence levels, requiring little computational efforts. We found that the difference in certifiable robustness for alpha=0.05 and alpha=0.0001 is already extremely small when drawing just 2,000 Monte-Carlo samples (Figure 20). Drawing 2,000 samples takes only 12 seconds on Cora-ML on average. This is significantly faster compared to all previous probabilistic certificates for GNNs that use up to 10^6 Monte-Carlo samples [2].
>
> ### What is the certification accuracy for different levels of alpha?
> In additional experiments with smaller confidence levels, we also found that the classification accuracy is high for just a few thousand Monte-Carlo samples. Please find more exhaustive experiments regarding accuracy for different levels of alpha and Monte-Carlo samples in Figure 21 (Appendix M).
>
> ### References
> [1] Alexander Levine, Soheil Feizi: Robustness Certificates for Sparse Adversarial Attacks by Randomized Ablation. AAAI 2020.
>
> [2] Aleksandar Bojchevski, Johannes Gasteiger, Stephan Günnemann. Efficient Robustness Certificates for Discrete Data: Sparsity-Aware Randomized Smoothing for Graphs, Images and More. ICML 2020.

---

> > ### Comment · Reviewer_dqSE · 2022-08-08
> > **Thank you for the additional results**
> >
> > Thank you for providing results for different levels of alpha. Although I do have concerns about this kind of probabilistic guarantee (as there is always a non-negligible chance the result is wrong), I understand this is also what other people claimed and not an issue of this single paper. Please do add these discussions to the paper. I keep my positive score and support the acceptance of this paper. Thank you.

---

> > > ### Author Response · Authors · 2022-08-08
> > > **Response to Reviewer dqSE**
> > >
> > > Thank you for your response!
> > >
> > > Although the confidence level $\alpha$ has just a minor effect on the strength of our certificates, we agree with your concerns regarding probabilistic certificates. Please note that there are already first works that "derandomize" probabilistic certificates [1,2]. The main idea is to evaluate the smoothed classifier exactly, which leads to deterministic certificates.
> > >
> > > One naive approach would be to simply iterate over the support of the smoothing distribution (all possible edge deletions and node ablations). However, similar to how we use the message-passing structure for certification, we can also leverage it to partition the support of the smoothing distribution into a smaller number of equivalence classes. While derandomization is not the focus of our work - and in fact orthogonal to the task of certification -  we will conduct additional derandomization experiments and add a discussion of this approach to the camera-ready version. We believe that future work can build upon it towards even more efficient and scalable derandomization schemes.
> > >
> > > Thank you.
> > >
> > > ### References
> > > [1] Alexander Levine, Soheil Feizi: (De)Randomized Smoothing for Certifiable Defense against Patch Attacks. NeurIPS 2020.
> > >
> > > [2] Alexander Levine, Soheil Feizi: Improved, Deterministic Smoothing for L1 Certified Robustness. ICML 2021.

---

### Official Review · Reviewer_KAVi · 2022-07-18

**Rating:** 7
**Confidence:** 4
**Soundness:** 4 excellent
**Presentation:** 4 excellent
**Contribution:** 3 good

**Summary:**

This paper proposes a novel method based on randomized smoothing to achieve robustness for graph neural networks. The task here is to certify the classification accuracy of GCN under any possible attacker who can perturb a bounded number of node features. Compared to the existing method, the proposed one considers random edge ablation and random node-level ablation. As a result, the approach achieves higher certified accuracy than existing baselines by a large margin.

**Questions:**

In terms of suggestions for the authors, see weaknesses.
Questions on writing:
- Line 234 and Appendix I: in the training process, how is the node mask t optimized?
- Line 284: what does $\epsilon=0.022$ stand for?
Suggestions on writing:
- Line 252-262: from the equation formulation, it is straightforward to see that the proposed certification is independent of perturbed feature dimensions $d$. Maybe we can omit Figure 4(a) and Line 252-262 to the appendix since it consumes much space.

**Limitations:**

See weaknesses.

**Strengths And Weaknesses:**

Strengths:
- Solid technical contributions: To the best of my knowledge, this is the first paper that applies random **edge** ablation. Existing approaches for certifying the smoothed GCNs only consider random **node** ablation. When edge ablation comes into play, the certification can take the graph topology into consideration to achieve a better robustness certification. This paper studies this problem in depth, by providing an upper bound of $\Delta$ for certification and characterizing the computing methods for exact certification for general graphs or tree-structured graphs.

- Great writing quality: the paper is easy to follow with just few typos/ambiguities. Especially, the paper studies the proposed smoothing scheme in-depth, beyond just showing a good empirical performance. This in-depth investigation includes theoretically characterizing the limitations and advantages of different graph smoothing protocols, proposing more expensive though tighter computing methods, and presenting quite a few ablation studies, especially those revealing the benefits of sparsity.

Weaknesses:
- The proposed method is a bit specific to graph neural networks with node perturbations. In other words, the method could be further benefited from extensions to a wider range of threat models, i.e., node addition/removel, edge feature perturbations, etc.

- More training techniques can be applied to further strengthen the certified robustness guarantee. For example, SmoothMix training (Jeong et al), SmoothAdv training (Salman et al), and DRT training (Yang et al).

Overall, I think these weaknesses are overall not critical. Disclaimer: I'm not actively working with GNNs. So I didn't evaluate the impact of this paper to GNN community nor whether the evaluation follows standardized protocols. I will refer to other reviewers' comments on these aspects to adjust my score if necessary.

---

> ### Author Response · Authors · 2022-08-02
> **Response to Reviewer KAVi**
>
> Thank you for your review!
>
> ### Concerning further extensions of our threat model (Comment 1)
> Thank you for pointing out that our method can be extended to a wider range of threat models. Please consider that we already discuss extensions like adversarial node insertion and deletion in Lines 206-210, where we point out that evaluating such certificates may not be possible for general graphs without further assumptions about how inserted/deleted nodes are connected to target nodes. Moreover, there are already robustness certificates for adversarial edge modification [1] (see also our response to Reviewer EH8E). We believe that sharing our core contribution is more important than introducing all its possible extensions, but we will discuss future work including such extensions more carefully in the camera-ready version.
>
> ### Concerning more training techniques (Comment 2)
> We agree with you that other training techniques may yield even stronger models in terms of accuracy and robustness. We will experiment with more training techniques (including the ones you mentioned) until the camera-ready deadline. Beyond that, we believe our main contribution to the scientific community is our robustness certificate, which (1) holds independently of the training technique, and (2) may further promote novel architectures and training methods towards more robust GNNs in the future.
>
> ### How is the node mask $t$ optimized?
> During training, we sample one different graph each epoch, where each sampled graph contains nodes with features replaced by the ablation representation $t$. For optimization, we implement $t$ as a parameter of our models: We initialize $t$ using Xavier initialization and we optimize it as we optimize the GNN weights during training. We added the corresponding details to Appendix I to better describe the optimization process, and also note that we uploaded our code as supplemental material.
>
> ### What does $\epsilon$=0.022 stand for?
> Graph Diffusion Convolution (GDC) [2] is a graph preprocessing technique that we use to sparsify the graph (Lines 274-284). We set the hyperparameter $\epsilon$ of GDC to 0.022, which controls the sparsification strength of the preprocessing technique. We consider this detail as an important hyperparameter for reproducing our result that sparsification improves certifiable robustness. We agree that this was unclear and we therefore added clarifying words.
>
> ### Lines 252-262 and Figure 4a)
> Thank you for pointing out potential for saving space, and we will consider this for the camera-ready version. We further believe that Lines 252-262 are important to discuss the difference between our method and existing certificates for GNNs.
>
> ### References
> [1] Aleksandar Bojchevski, Johannes Gasteiger, Stephan Günnemann. Efficient Robustness Certificates for Discrete Data: Sparsity-Aware Randomized Smoothing for Graphs, Images and More. ICML 2020.
>
> [2] Johannes Gasteiger, Stefan Weißenberger, Stephan Günnemann: Diffusion Improves Graph Learning. NeurIPS 2019.

---

> > ### Comment · Reviewer_KAVi · 2022-08-06
> > **Thank you for the response**
> >
> > Thanks for the authors for the clear response and efforts on paper revision. Most of my concerns are addressed. There is just one concern remaining:
> >
> > - What is the model's training $p_d$ and $p_a$ throughout the experimental evaluation? The manuscript does not mention that they use node or edge ablation in "Section 6 - Datasets and models'' paragraph. So I am confused on whether node and edge ablation are actually in use during training. Though by inferring from the context like training of $t$ mask or "different $p_d$ and $p_a$ during training and inference" I can know $p_d$ and $p_a$ ablations are used during training, this point seems not be made clear, and the actually values of $p_d$ and $p_a$ in training are either omitted or not shown in an apparent place.

---

> > > ### Author Response · Authors · 2022-08-08
> > > **Response to Reviewer KAVi**
> > >
> > > Thank you for your follow-up question!
> > >
> > > We also delete edges and ablate node features during training (using different probabilities $p_d$ for edge deletion and $p_a$ for node feature ablation during training and inference). Please note that only the probabilities at inference eventually determine the strength of our certificates, while the probabilities used during training are just hyperparameters that we can optimize to improve the robustness-accuracy trade-offs. We consider this in our extensive experiments (Lines 297-304), where we explore the joint space of training-time and inference-time smoothing parameters.
> > >
> > > In response to your comment, we now explicitly state in the paper that we also delete edges and ablate node features during training (Lines 223-224). In addition, we specify the values of $p_d$ and $p_a$ that we use to train specific models (see added Appendix N). Moreover, we will also consider your comment for improving our camera-ready version by adding additional experiments on the effect of choosing different $p_d$ and $p_a$ during training.
> > >
> > > Thank you.

---

> > > > ### Comment · Reviewer_KAVi · 2022-08-08
> > > > **Response**
> > > >
> > > > Thank you for the clarification and updating the paper. I am satisfied with the response. Therefore, I decided to keep my support of acceptance of this paper.

---

### Author Response · Authors · 2022-08-02
**Overview Comment**

We replied to all reviewers and changed our initial submission in response to their comments as follows:

- Reviewer KAVi: Additional optimization details in Appendix I
- Reviewer KAVi: Additional training details (training-time smoothing parameters) in Appendix N
- Reviewer dqSE: Additional experiments using smaller confidence levels in Appendix M
- Reviewer 51To: Additional motivation for our threat model in Appendix L

---

### Meta-Review · Area_Chair_9zw8 · 2022-08-25

**Recommendation:** Accept
**Confidence:** Certain

**Metareview:**

The paper proposes a novel approach to certify the robustness of graph neural networks via randomized smoothing. It does so by treating the networks as ``gray-box'' models and leveraging message passing routines. This yields an improved lower bound on probabilistic certification.

All reviewers recognized the technical quality of the work and its novel perspective on adversarial robustness for graph neural networks. Some concerns regarding the probabilistic certification and the experimental details have been successfully addressed during the rebuttal.

The paper is recommended for acceptance, conditioned on the inclusion of the additional experiments and discussions arisen in the rebuttal.

**Award:**

No

---

### Decision · Program_Chairs · 2022-09-14

Accept